# COMPUTING LOW-ENTROPY COUPLINGS FOR LARGE-SUPPORT DISTRIBUTIONS

## ABSTRACT

A minimum-entropy coupling is a joint probability distribution having minimum joint entropy among all joint distributions with given pre-specified marginals. While provable approximation algorithms for a minimum-entropy coupling exist, they take log-linear time in the size of the support of the marginal distributions. Thus, applications involving very large-support distributions instead use a class of heuristic iterative minimum-entropy coupling (IMEC) algorithms. Unfortunately, existing IMEC algorithms are limited to specific classes of distributions, prohibiting applications involving general large-support distributions. In this work, we resolve this issue by making three main contributions: 1) We unify existing IMEC algorithms under a single formalism using sets of partitions. 2) We derive a new IMEC instance from this formalism, which we call ARIMEC, that, unlike existing IMEC algorithms, can be applied in practice to arbitrary discrete distributions, and introduce associated operations that make ARIMEC efficient in practice. 3) We empirically show the utility of ARIMEC for both Markov coding games and steganography.

## 1 INTRODUCTION

Given two marginal distributions, a coupling is a bivariate joint distribution with the given marginals. In general, there may be many couplings for a particular pair of marginals. The problem of computing a coupling with the minimum amount of joint entropy among all feasible couplings, given access to probability mass evaluations of the marginals, is called minimum-entropy coupling (MEC) (Kovačević et al., 2015). As is detailed in (Compton et al., 2023), applications of MEC include causal inference (Kocaoglu et al., 2017; Compton et al., 2020; Javidian et al., 2021; Compton et al., 2022), communication (Sokota et al., 2022), steganography (Schroeder de Witt et al., 2023), random number generation (Li, 2021), functional representations (Cicalese et al., 2019), and dimensionality reduction (Vidyasagar, 2012; Cicalese et al., 2016).

While MEC is NP-hard (Kovačević et al., 2015), recent works have provided approaches that achieve provable approximations of MECs (Kocaoglu et al., 2017; Cicalese et al., 2019; Rossi, 2019; Li, 2021; Compton et al., 2023) in log-linear time (i.e., $O(N \log N)$) in the cardinality of the support of the marginals. Unfortunately, the supports of many distributions of practical interest, such as those of deep generative models, are intractably large for these provable approximation algorithms.

To handle such cases, Sokota et al. (2022) introduced a class of heuristic algorithms for producing low-entropy couplings. These algorithms work by iteratively coupling components of random vectors using provable MEC approximation algorithms in such a way that guarantees the aggregate joint distribution is a coupling. In practice, both Sokota et al. (2022) and Schroeder de Witt et al. (2023) find that these iterative minimum-entropy coupling (IMEC) approaches produce low-entropy couplings for distributions with very large supports—binary images and trajectories of Atari games (Bellemare et al., 2013) in the work of Sokota et al. (2022) and binary strings and generative models (including GPT-2 (Radford et al., 2019a), WaveRNN (Kalchbrenner et al., 2018), and Image Transformer (Parmar et al., 2018)) in the work of Schroeder de Witt et al. (2023). Unfortunately, the applicability of the IMEC algorithms Sokota et al. (2022) introduced is limited to problems in which one distribution either has small support or is factorable. As a result, at the time of writing, there are no techniques for producing low-entropy couplings of general large-support distributions.

In this work, we make multiple contributions regarding the IMEC line of research. First, we unify existing IMEC algorithms under a single formalism using *sets of partitions*. In this unified perspec-

tive, IMEC couples distributions with intractably large supports by performing (approximate) MECs between autoregressive distributions of one marginal and the posterior over the blocks of a partition of the support of the other marginal. At a particular iteration, IMEC uses a partition that maximizes entropy over the blocks of that partition. Within this formalism, the two existing IMEC algorithms can be viewed as using sets of partitions that include every partition and that include only partitions corresponding to each component of the factorization, respectively.

Leveraging this formalism, we derive the first algorithm for computing low-entropy couplings for arbitrary large-support distributions, which we call ARIMEC. ARIMEC uses a set of partitions, which we refer to as the prefix tree partition set, in which each partition corresponds to a node of the prefix tree of the support of a marginal distribution. These prefix trees can have very large numbers of nodes (and therefore induce very large numbers of partitions). Thus, to a facilitate an efficient implementation, we introduce techniques to 1) lazily update the posterior over different blocks and 2) upper bound the entropy induced by the partitions in large parts of the prefix tree.

We demonstrate ARIMEC's utility empirically in both Markov coding games (Sokota et al., 2022)—a setting in which the objective is to encode messages into the trajectories of Markov decision process while simultaneously achieving a large expected return—and steganography (Cachin, 1998)—a setting in which the objective is to encode (sensitive) information into innocuous-seeming content in such a way that an adversary would not realize that there is hidden information. We obtain significantly improved communication rates in both settings, illustrating how ARIMEC is uniquely able to leverage autoregressive prior information about realistic messages.

## 2 BACKGROUND AND NOTATION

For our background, we formally introduce minimum-entropy coupling and discuss existing techniques for computing and (heuristically) approximating minimum-entropy couplings. We also discuss the conditions (and their limitations) required for these existing algorithms to be computationally tractable. Thereafter, we introduce notation for partitions of sets, which we will use to unify existing methods in one general framework.

### 2.1 MINIMUM-ENTROPY COUPLING

We begin by formalizing the ideas of the idea of couplings and minimum entropy couplings.

**Definition 2.1.** *Let $\mu\colon \mathbb{X} \to [0,1]$ be a probability distribution over a finite set $\mathbb{X}$ and let $\nu\colon \mathbb{Y} \to [0,1]$ be a probability distribution over a finite set $\mathbb{Y}$. A **coupling** of $\mu$ and $\nu$ is a bivariate joint probability distribution $\gamma\colon \mathbb{X} \times \mathbb{Y} \to [0,1]$ that marginalizes to $\mu$ and $\nu$. In other words, $\gamma$ satisfies*

$$\sum_{x' \in \mathbb{X}} \gamma(x', y) = \nu(y) \text{ for all } y \in \mathbb{Y} \text{ and } \sum_{y' \in \mathbb{Y}} \gamma(x, y') = \mu(x) \text{ for all } x \in \mathbb{X}. \tag{1}$$

*We use $\Gamma(\mu, \nu) = \{\gamma \mid \gamma \text{ satisfies (1)}\}$ to denote the set of all couplings for $\mu$ and $\nu$.*

**Definition 2.2.** *Given a coupling $\gamma$, the **joint entropy** is defined as $\mathcal{H}(\gamma) = -\mathbb{E}_{(X,Y)\sim\gamma} \log \gamma(X, Y)$.*

Throughput the paper, we will use capital letters to denote random variables, as is done above.

**Definition 2.3.** *Given two marginal distributions $\mu, \nu$, a **minimum-entropy coupling** is a coupling $\gamma \in \Gamma(\mu, \nu)$ such that $\mathcal{H}(\gamma) = \min\{\mathcal{H}(\gamma') \mid \gamma' \in \Gamma(\mu, \nu)\}$.*

### 2.2 COMPUTING AND APPROXIMATING MINIMUM-ENTROPY COUPLINGS

While computing an exact minimum-entropy coupling is NP-hard (Kovačević et al., 2015), there has been a series of recent works that prove guarantees for approximate minimum-coupling algorithms running in $O(N \log N)$ time, where $N$ is the cardinality of the support of the marginals. Cicalese et al. (2019) introduced an approximation algorithm that they show guarantees a coupling within 1 bit of minimum entropy. Rossi (2019) showed that Kocaoglu et al. (2017)'s greedy approach also guarantees a coupling within 1 bit of minimum entropy. Li (2021) introduced a third approach for which he also proved a 1 bit approximation guarantee. Most recently, Compton et al. (2023) showed an improved guarantee for Kocaoglu et al. (2017)'s greedy approach of about 0.53 bits, while also showing that Cicalese et al. (2019) and Li (2021)'s algorithms cannot match this guarantee. Compton et al. (2023) also give approaches that guarantee exact MECs, though they require exponential time.

## 2.3 Iterative Minimum-Entropy Coupling with a Tabular Posterior

In some settings, it is desirable to (non-provably) approximate minimum-entropy couplings where one random vector assumes such a large number of possible outcomes that the approaches described in Section 2.2 are inapplicable. Sokota et al. (2022) propose an iterative approach to such settings, which we call TIMEC, that assumes the random vector is autoregressively specified. TIMEC guarantees that the resulting joint distribution is a coupling, supports conditional sampling and likelihood queries for both $X \mid Y$ and $Y \mid X$, where $Y$ is the random vector, and heuristically achieves low entropy. It can either be defined using the conditional generative process for sampling $Y$ given $X$ or the conditional generative process for sampling $X$ given $Y$, as both induce the same joint distribution. We focus on the process for generating $Y$ given $X$, which is formalized in Algorithm 1, in the main body but include the process for generating $X$ given $Y$ in Algorithm 4 in Appendix A. Algorithm 1 works iteratively in two steps:

1. First it performs an (approximate) MEC between the (inductively defined) posterior over $X$ given $Y_{1:j-1}$ (inductively defined via Bayes' Theorem) and the conditional distribution $\nu(Y_j \mid Y_{1:j-1})$. The joint posterior over $X$ and $Y_j$ given $Y_{1:j-1}$ is assigned to the output of this coupling.

2. Second, it samples $Y_j$ from the posterior over $Y_j$ given both $X = x$ and $Y_{1:j-1}$ (also inductively defined via Bayes' Theorem).

Note that we use upper-bound-inclusive indexing, so $Y_{1:0} = ()$, $Y_{1:1} = (Y_1)$, $Y_{1:2} = (Y_1, Y_2)$, etc.

---

**Algorithm 1** TIMEC: $Y \mid X = x$

---

**procedure** TIMEC($\mu, \nu, x$)
    $\gamma(X) \leftarrow \mu(X)$
    **for** $j = 1, \ldots, m$ **do**
        $\gamma(X, Y_j \mid Y_{1:j-1}) \leftarrow \text{MEC}(\gamma(X \mid Y_{1:j-1}), \nu(Y_j \mid Y_{1:j-1}))$
        $Y_j \sim \gamma(Y_j \mid x, Y_{1:j-1})$
    **end for**
    return $Y$
**end procedure**

---

TIMEC has the following runtime guarantee.

**Proposition 2.1** (TIMEC Runtime)**.** *Algorithm 1 can be implemented in $O(m \max(M, |\mathbb{X}|) \log \max(M, |\mathbb{X}|))$ time, where $M = \max_j |\mathbb{Y}_j|$.*

We prove Proposition 2.1 in Appendix B.1. Thus, TIMEC is polynomial in $m, |\mathbb{X}|, |\mathbb{Y}_1|, \ldots, |\mathbb{Y}_m|$. Note that is in contrast to a direct application on an approximate MEC algorithm, which would require $O(\max(M^m, |\mathbb{X}|) \log \max(M^m, |\mathbb{X}|))$ time, using the same notation.

## 2.4 Iterative Minimum-Entropy Coupling with a Factored Posterior

Unfortunately, having polynomial cost in $|\mathbb{X}|$ makes TIMEC inapplicable to many settings, such as steganography with large message sizes (Schroeder de Witt et al., 2023). To ameliorate this issue, Sokota et al. (2022) also proposed a second approach, which we call FIMEC[1], in which $X$ is also assumed to be a random vector. Furthermore, crucially, it is assumed to be factorable, as is formalized below.

**Assumption 2.4.** *$X = (X_1, \ldots, X_n)$ is a random vector such that $X_i$ and $X_j$ are independently distributed for $i \neq j$.*

As with TIMEC, FIMEC guarantees that the resulting distribution is a coupling, supports likelihood queries to both conditionals and the joint distribution, and heuristically achieves low entropy. It can similarly be defined in terms of either conditional generative process ($X \mid Y$ or $Y \mid X$). We again focus on the $Y \mid X$ case (Algorithm 2), and defer the $X \mid Y$ case to Appendix A. The basic structure of Algorithm 2 is analogous to that of Algorithm 1. However, rather than performing MECs using $\gamma(X \mid Y_{1:j-1})$, FIMEC uses $\gamma(X_{i^*} \mid Y_{1:j-1})$, where $X_{i^*}$ is a component of $X$ with maximum posterior entropy. The other components $X_i$ for $i \neq i^*$ are left independent of $Y_j \mid Y_{1:j-1}$.

---

[1] Note that Schroeder de Witt et al. (2023) use the name iMEC for this approach.

---

**Algorithm 2** FIMEC: $Y \mid X{=}x$

---

  **procedure** FIMEC($\mu, \nu, x$)
      $\gamma(X) \leftarrow \mu(X)$
      **for** $j = 1, \ldots, m$ **do**
         $i^* \leftarrow \operatorname{argmax}_i \mathcal{H}(\gamma(X_i \mid Y_{1:j-1}))$
         $\gamma(X_{i^*}, Y_j \mid Y_{1:j-1}) \leftarrow \text{MEC}(\gamma(X_{i^*} \mid Y_{1:j-1}), \nu(Y_j \mid Y_{1:j-1}))$
         $\gamma(X, Y_j \mid Y_{1:j-1}) \leftarrow \gamma(X_{i^*}, Y_j \mid Y_{1:j-1}) \cdot \left( \prod_{i \neq i^*} \gamma(X_i \mid Y_{1:j-1}) \right)$
         $Y_j \sim \gamma(Y_j \mid x, Y_{1:j-1})$
      **end for**
      return $Y$
  **end procedure**

---

FIMEC has the following runtime guarantee.

**Proposition 2.2** (FIMEC Runtime). *Let Assumption 2.4 hold. Then Algorithm 2 can be implemented in $O(m \max(M, N) \log \max(M, N) + nN + m \log n + n \log n)$ time, where $N = \max_i |\mathbb{X}_i|$ and $M = \max_j |\mathbb{Y}_j|$.*

We prove Proposition 2.2 in Appendix B.1. Thus, under Assumption 2.4, FIMEC is polynomial in $m, n, |\mathbb{X}_1|, \ldots, |\mathbb{X}_n|, |\mathbb{Y}_1|, \ldots, |\mathbb{Y}_m|$. Note again that this is in contrast to a direct application of an approximate MEC algorithm, which would require $O(\max(N^n, M^m) \log \max(N^n, M^m))$ time, using the same notation.

### 2.5 PARTITIONS OF SETS

As discussed in the introduction, we will show the IMEC algorithms discussed in the previous two sections can be unified into a single algorithm using partitions over the possible values of $X$. We use the following definitions and notation for partitions.

**Definition 2.5.** *A partition $\mathcal{P}$ of a set $\mathbb{X}$ is a set of blocks $\{\mathbb{B}_1, \ldots, \mathbb{B}_\ell\}$ where: 1) Each block $\mathbb{B}_k \in \mathcal{P}$ is a subset of $\mathbb{X}$; 2) Every pair of blocks $\mathbb{B}_k, \mathbb{B}_{k'} \in \mathcal{P}$ has an empty intersection; 3) The union of blocks $\cup_{k=1}^{\ell} \mathbb{B}_k$ is equal to $\mathbb{X}$.*

**Definition 2.6.** *For a partition $\mathcal{P}$ of a set $\mathbb{X}$, the block function $\mathcal{B}_\mathcal{P} : \mathbb{X} \to \mathcal{P}$ maps $x$ to the block of the partition of which it is an element.*

When $X$ is a random variable, we use $B_\mathcal{P} = \mathcal{B}_\mathcal{P}(X)$ to denote the block of $\mathcal{P}$, as a random variable, to which $X$ belongs. Note that a block's probability is the sum of the probabilities of its elements.

## 3 A UNIFICATION OF ITERATIVE MINIMUM-ENTROPY COUPLING

We are now ready to describe our unification of existing IMEC algorithms. In this unification, different instances of IMEC are specified using different sets of partitions $\mathfrak{U} \subset \{\mathcal{P} \mid \mathcal{P} \text{ is a partition of } \mathbb{X}\}$. Both existing and any new instances of this unified perspective guarantee that the resulting distribution is a coupling, support conditional and likelihood queries for both $X \mid Y$ and $Y \mid X$, and heuristically produce low entropy. We define this unified perspective to IMEC using the conditional generative process given in Algorithm 3, which samples from $Y|X$. (Equivalently, it is defined by the generative process given in Algorithm 6 in Appendix A, which samples from $X|Y$.) Algorithm 3 works iteratively in three steps:

1. First, it computes the partition $\mathcal{P} \in \mathfrak{U}$ inducing posterior maximum entropy. The entropy induced by a partition $\mathcal{P}$ at iteration $j$ is defined in terms of the probabilities over the blocks of the partition under $\gamma$, given $Y_{1:j-1}$. That is, $\mathcal{H}(\gamma(B_\mathcal{P} \mid Y_{1:j-1})) = -\sum_{\mathbb{B} \in \mathcal{P}} \gamma(\mathbb{B} \mid Y_{1:j-1}) \log \gamma(\mathbb{B} \mid Y_{1:j-1})$. The intuition behind selecting the maximum-entropy partition is that it heuristically offers the opportunity to reduce the joint entropy by the largest amount.[2]

---

[2]A justification is as follows. Recall that $\max(\mathcal{H}(C), \mathcal{H}(D)) \leq \mathcal{H}(C, D) \leq \mathcal{H}(C) + \mathcal{H}(D)$, where $\mathcal{H}(C, D)$ achieves its upper bound when $C$ and $D$ are independent. Thus, the maximum reduction in joint entropy achievable by performing a coupling is upper bounded by $\mathcal{H}(C) + \mathcal{H}(D) - \max(\mathcal{H}(C), \mathcal{H}(D)) = \min(\mathcal{H}(C), \mathcal{H}(D))$. Therefore, maximizing $\mathcal{H}(C)$ maximizes an upper bound on the joint entropy reduction.

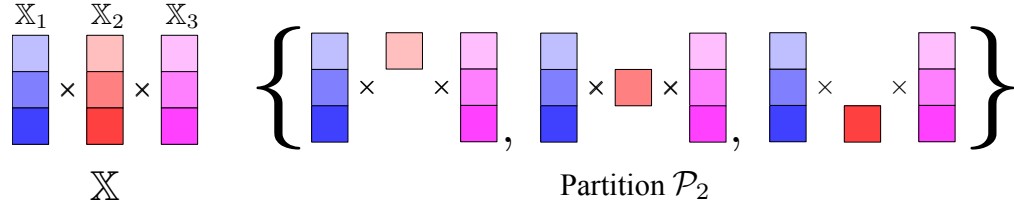

Figure 1: (Left) A set $\mathbb{X}$ of sequences of length 3; (right) the partition $\mathcal{P}_2$ used in FIMEC.

2. Second, it performs an (approximate) MEC between the posterior over the block of the chosen partition $B_{\mathcal{P}}$ given $Y_{1:j-1}$ and the conditional distribution $\nu(Y_j \mid Y_{1:j-1})$. The joint posterior over the block $B_{\mathcal{P}}$ and $Y_j$ given $Y_{1:j-1}$ is assigned to the output of this coupling.

3. Third, it samples $Y_j$ from the posterior over $Y_j$ given both the true block $\mathcal{B}_{\mathcal{P}}(x)$ of $X$ and $Y_{1:j-1}$.

---

**Algorithm 3** IMEC (Generic Form): $Y \mid X = x$

---
    **procedure** IMEC($\mu, \nu, x, \mathfrak{U}$)
        $\gamma(X) \leftarrow \mu(X)$
        **for** $j = 1, \ldots, m$ **do**
            $\mathcal{P} \leftarrow \arg\max_{\mathcal{P} \in \mathfrak{U}} \mathcal{H}(\gamma(B_{\mathcal{P}} \mid Y_{1:j-1}))$
            $\gamma(B_{\mathcal{P}}, Y_j \mid Y_{1:j-1}) \leftarrow \text{MEC}(\gamma(B_{\mathcal{P}} \mid Y_{1:j-1}), \nu(Y_j \mid Y_{1:j-1}))$
            $Y_j \sim \gamma(Y_j \mid \mathcal{B}_{\mathcal{P}}(x), Y_{1:j-1})$
        **end for**
        return $Y$
    **end procedure**

---

Note that whether Algorithm 3 can be implemented efficiently depends on the distribution $\mu$ and the set of partitions $\mathfrak{U}$.

## 3.1 THEORY

The general form of IMEC possesses the following two properties, which reduce to the results of Sokota et al. (2022) as a special case.

**Proposition 3.1** (Coupling). *IMEC induces a coupling of $\mu$ and $\nu$.*

**Proposition 3.2** (Greediness). *If the trivial partition (i.e., the partition of singletons) is in $\mathfrak{U}$, IMEC approximately minimizes $\mathcal{H}(X, Y_{1:j})$ subject to $\mu, \nu, \gamma(X, Y_{1:j-1})$ on the $j$th iteration, for each $j$.*

Proofs for these statements are provided in Appendix B.2 and Appendix B.3, respectively.

## 3.2 SPECIAL CASES

**Tabular Posterior** To implement TIMEC using Algorithm 3, we can select the partition set $\mathfrak{U}$ to be the set of all partitions of $\mathbb{X}$. As per Lemma B.1, which we state and prove in Appendix B.4, the trivial partition (or one that is equivalent up to measure zero) will always be selected as it achieves maximum-entropy. Coupling with the trivial partition is equivalent to coupling over the whole set, which is exactly what TIMEC does.

**Factored Posterior** To implement FIMEC using Algorithm 3 we can select the partition set as $\mathfrak{U} = \{\mathcal{P}_1, \ldots, \mathcal{P}_n\}$, where for each $i$, $\mathcal{P}_i = \{\mathbb{X}_1 \times \cdots \times \mathbb{X}_{i-1} \times \{x_i\} \times \mathbb{X}_{i+1} \times \cdots \times \mathbb{X}_n \mid x_i \in \mathbb{X}_i\}$ and where $\mathbb{X}_i$ denotes the set of possible values for $X_i$. An example is shown in Figure 1. From this perspective, selecting $\mathcal{P}_i$ on a particular iteration is equivalent to selecting $X_i$ as the component with which to couple.

## 4 A GENERAL APPROACH TO ITERATIVE MINIMUM-ENTROPY COUPLING

In this section, building on our unified framework, we derive a general IMEC algorithm, which we call ARIMEC. ARIMEC improves upon the generality of FIMEC by dropping the Assumption 2.4, which does not hold in general. We present ARIMEC in two parts. First, we introduce the *prefix tree partition set*, which allows us to formally define ARIMEC using Algorithm 3. Second, we highlight the insights required to efficiently implement ARIMEC, which are detailed fully in the appendix.

### 4.1 THE PREFIX TREE PARTITION SET

In order to define the prefix tree partition set, we first define prefixes.

**Definition 4.1.** *We write $v \sqsubset v'$ to mean that $v$ is a prefix of $v'$ in the substring sense and, equivalently, that $v'$ is an extension of $v$ in the substring sense.*

**Definition 4.2.** *We say $v$ is the immediate prefix of $v'$ and, equivalently, that $v'$ is the immediate extension of $v$ if $v \sqsubset v'$ and $v'$ is one character longer than $v$.*

Next, we define the *prefix tree* of a set of vectors as the following directed graph.[3]

**Definition 4.3.** *The prefix tree for a set of vectors $\mathbb{X}$ is a directed graph $(\mathbb{V}, \mathbb{E})$, where the vertex set $\mathbb{V} = \{v \sqsubset x \mid x \in \mathbb{X}\}$ is the set of prefixes of elements of $\mathbb{X}$ and the set of edges $\mathbb{E} = \{(v, c) \mid v, c \in \mathbb{V}, v$ is the immediate prefix of $c\}$ is the set of pairs of vertices and their immediate prefixes.*

The prefix tree induces a set of partitions over $\mathbb{X}$ where each partition corresponds to a tree node. We call this set of partitions, defined below, the prefix tree partition set.

**Definition 4.4** (Prefix tree partition set). *Let $(\mathbb{V}, \mathbb{E})$ be the prefix tree for $\mathbb{X}$. Then the prefix tree partition set is defined as $\mathfrak{U} = \{\mathcal{P}_v \mid v \in \mathbb{V}\}$, where*

$$\mathcal{P}_v = \{\mathbb{B}_{c\sqsubset} \mid (v, c) \in \mathbb{E}\} \cup \{\mathbb{B}_{v\not\sqsubset}\} \cup \{\mathbb{B}_{v=}\},$$

*where $\mathbb{B}_{c\sqsubset} = \{x \in \mathbb{X} \mid c \sqsubset x\}$ denotes the subset of $\mathbb{X}$ that is an extension of the child $c$, where $\mathbb{B}_{v\not\sqsubset} = \{x \in \mathbb{X} \mid v \not\sqsubset x\}$ denotes the subset of $\mathbb{X}$ that does not extend $v$, and where $\mathbb{B}_{v=} = \{x \in \mathbb{X} \mid v = x\}$ denotes the (either singleton or empty) subset of $\mathbb{X}$ equal to $v$.*

In the prefix tree partition set, there is a partition $\mathcal{P}_v$ for each node $v$ in the prefix tree. For the partition $\mathcal{P}_v$, for each child of $v$, all elements of $\mathbb{X}$ prefixed by that child constitute a block; all elements of $\mathbb{X}$ that are not prefixed by $v$ constitute a block; and if $v$ is itself a member of $\mathbb{X}$, it occupies an additional singleton block. A visualization of the prefix tree and one partition that it induces is shown in Figure 2.

### 4.2 ITERATIVE MINIMUM-ENTROPY COUPLING WITH AN AUTOREGRESSIVE POSTERIOR

Having defined the prefix tree partition set, we can now formalize ARIMEC.

**Definition 4.5** (ARIMEC). *ARIMEC is the instance of Algorithm 3 in which the set of partitions $\mathfrak{U}$ is selected to be the prefix tree partition set.*

To provide more grounded intuition for ARIMEC, Appendix C shows visualizations of example iterations of FIMEC in Figure 6 and ARIMEC in Figure 7 for marginals of length two. ARIMEC possesses the following runtime guarantee.

**Proposition 4.1** (ARIMEC Runtime). *Algorithm 3 with $\mathfrak{U}$ set to the prefix tree partition set runs in $O(m \max(M, N) \log \max(M, N) + mZN)$, where $N = \max_i |\mathbb{X}_i|$ and $M = \max_j |\mathbb{Y}_j|$ and where $Z$ is (a function of $n$ and $N$) that denotes the number of nodes in the prefix tree that require checking to find the maximum-entropy partition.*

We prove Proposition 4.1 in Appendix B.1. Assuming that we do not need to check a large number of nodes to find the maximum-entropy partition, Proposition 4.1 guarantees efficient runtime. However, under a naive implementation of ARIMEC $Z = N^n$. Thus, to facilitate the usage of ARIMEC in practice, we prove the following result.

---

[3]Note that our usage is graph theoretic and does not pertain to the trie data structure.

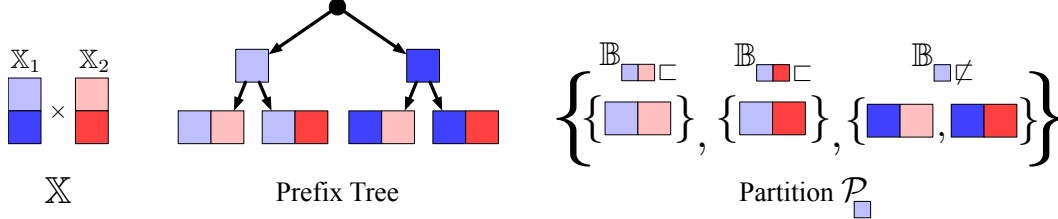

Figure 2: (Left) A set $\mathbb{X}$ of sequences of length 2; (middle) the prefix tree for $\mathbb{X}$; (right) the partition induced by the left-most depth one node of the prefix tree.

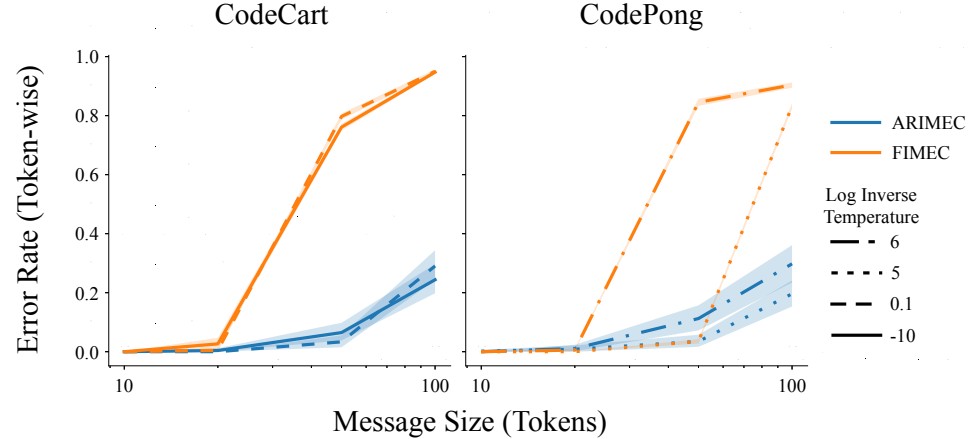

Figure 3: Results for the Markov coding games CodeCart and CodePong using MaxEntRL policies with different temperatures with 95% bootstrap confidence intervals drawn from 100 games.

**Proposition B.2** (Maximum-Entropy Partition). *(Informal) Fix any node $v$ in the prefix tree. Let $u$ be a neighbor of $v$. If $u$ is the parent of $v$, let $q = \gamma(\mathbb{B}_{u\not\sqsubset} \mid Y_{1:j})$; if $u$ is a child of $v$, let $q = \gamma(\mathbb{B}_{u\sqsubset} \mid Y_{1:j})$. Then, if $q < 1 - 1/N$, where $N = \max_i |\mathbb{X}_i|$, it follows that $\mathcal{H}(\gamma(B_{\mathcal{P}_{v'}} \mid Y_{1:j}) \leq -(1-q)\log(1-q) - q\log\frac{q}{1-N}$ for any node $v'$ such that the path between $v$ and $v'$ includes $u$.*

For outgoing edges with sufficiently small probabilities, Proposition B.2 allows us to upper bound the entropies of every partition associated with a subtree or complement of a subtree in the prefix tree. Thus, if we already have observed a partition with a higher entropy than the upper bound, the entire subtree (or subtree complement) can be pruned for the purposes of computing the maximum-entropy partition. We observed that Proposition B.2 facilitates very aggressive pruning; the average number of partitions per iteration that required checking ranged from less than one to slightly more than two across our experiments. Thus, although Proposition 4.1 does not give a polynomial time guarantee on ARIMEC, we find that it can be efficient in practice.

## 5 EXPERIMENTS

To demonstrate the effectiveness of ARIMEC, we perform experiments in two settings: Markov coding games (Sokota et al., 2022) and steganography (Cachin, 1998).

### 5.1 MARKOV CODING GAMES

In a Markov coding game (MCG) (Sokota et al., 2022), the goal is to communicate messages via the trajectories of a Markov decision process (MDP), while simultaneously achieving a high expected

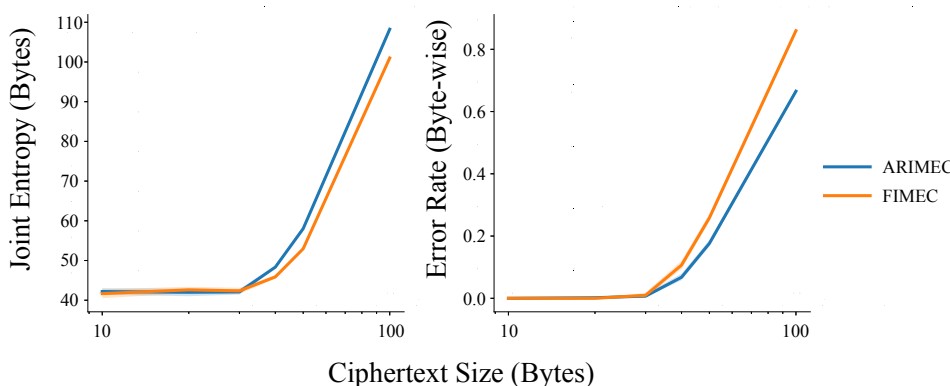

Figure 4: Results for information-theoretic steganography with 95% bootstrap confidence intervals drawn from 100 samples.

return. Messages are sampled independently from a distribution known to both the players sending and receiving them. For a more complete description, see Appendix D.1.

Sokota et al. (2022) give a principled approach to this setting called MEME that works in two steps. First, MEME trains a maximum-entropy reinforcement learning (MaxEntRL) (Ziebart et al., 2008) policy for the MDP. (The intuition is that this policy balances between performing well in the MDP and having high bandwidth through which information can be communicated.) Second, MEME computes (or approximates) a minimum-entropy coupling between the distribution over messages and, roughly speaking, the distribution over trajectories induced by the MaxEntRL policy.[4] MEME guarantees that the expected return in the MCG is the same as in the MDP; furthermore, at each time step, MEME greedily maximizes the amount of information encoded into the trajectory. For a more complete description, see Appendix D.2.

Because the second step of MEME requires computing or approximating a MEC, prior to this work, it was only applicable to MCGs whose message distributions had small or factorable supports. Thus our extension of IMEC to arbitrary distributions also serves as an extension of MEME to arbitrary MCGs. To illustrate the benefits of MEME's extended we perform experiments in two MCGs based on Cartpole and Pong (Bellemare et al., 2013), which we call CodeCart and CodePong, that were previously beyond MEME's applicability. For these MCGs, the distribution over messages is dictated by GPT-2 (Radford et al., 2019b) with top-50 sampling. For each game, we trained two policies with using different entropy bonus temperatures that each achieved perfect scores in 100/100 games. As a baseline, we compare against a naive version of MEME that assumes that the message was sampled from a uniform distribution over tokens and uses FIMEC. Note that this baseline sacrifices MEME's expected return guarantee.

We show the rate at which trajectories are decoded incorrectly for each variant of IMEC in these settings (Figure 3). While both FIMEC and ARIMEC maintain perfect expected return, ARIMEC produces a substantially more efficient encoding.

## 5.2 STEGANOGRAPHY

In steganography, the goal is to encode information (called plaintext) into innocuous-seeming content (called stegotext), such that an adversary would not realize that the innocuous-seeming content contains hidden information. We consider two kinds of steganography for our experiments. The first is information-theoretic steganography (Cachin, 1998), which seeks formal security guarantees. Schroeder de Witt et al. (2023) proved that this problem can be reduced to minimum-entropy coupling distributions of ciphertext (random bitstrings) with distributions of covertext (innocuous content). For a more complete description, see Appendix D.3.

---

[4]To be more precise about the latter distribution requires nuance since environment transitions cannot be correlated with the message. See Sokota et al. (2022) for details.

In this setting, Assumption 2.4 holds; thus, we would expect FIMEC to perform well relative to the ARIMEC. We show both the resulting joint entropy and the rate at which the ciphertext is decoded incorrectly in Figure 4, using 100 tokens sampled from GPT-2 as the covertext. This error rate can be written as $\mathbb{E}_{X \sim \mathcal{X}} \mathbb{E}_{Y \sim \gamma(Y|X)} I[X \neq \arg\max_x \gamma(x \mid Y)]$. Interestingly, while FIMEC produces lower joint entropy than ARIMEC, ARIMEC appears to produce lower decoding error. This could be because the ARIMEC focuses on maximizing the certainty of the bytes earlier in the string, while FIMEC focuses on reducing the uncertainty about the most uncertain bytes; thus, ARIMEC may be more likely to get at least some of the string correct.

The second kind of setting is unencrypted steganography. The unencrypted setting we consider differs in that the distribution over plaintext messages is known to the sender and receiver, and there is no private key exchange. Thus the sender is forced to encode the plaintext message directly into stegotext (rather than encrypting it into ciphertext). This setting has the advantages of higher potential information throughput (by leveraging prior information on the message distribution) and not requiring a secure channel for private key exchange, but the disadvantages of strong assumptions on the plaintext message distribution and weaker security guarantees. In Appendix D.4, we provide novel results showing that coupling-based approaches to this setting provide perfect undetectability (Theorem D.7) and that minimum-entropy coupling-based approaches provide the highest information throughput among perfectly undetectable approaches (Theorem D.8). Appendix D.4 also includes a more complete description of the problem setting and discussion of the advantages and disadvantages.

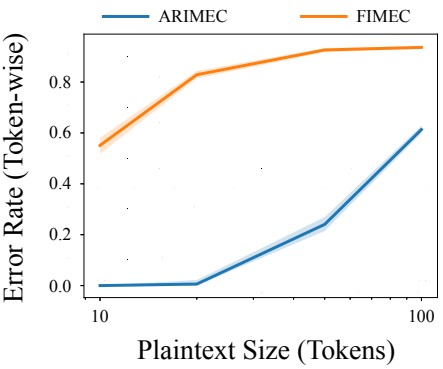

Figure 5: Results for unencrypted steganography with 95% bootstrap confidence intervals drawn from 100 samples.

To test ARIMEC, we perform experiments where the covertext distribution is dictated by 100 tokens sampled from GPT-2 with the prompt "Here's an innocuous message:" and the plaintext message distribution is dictated by GPT-2 with the prompt "Here's a secret message:". We compare ARIMEC (with the correct prior) against FIMEC that incorrectly assumes a uniform distribution over tokens. Note that the former maintains perfect undetectability guarantees, while the latter does not. We show the results of this experiment in Figure 5. Interestingly, we find that ARIMEC outperforms FIMEC in terms of information throughput to a much greater extent than in our MCG experiments.

## 6 CONCLUSION AND FUTURE WORK

In this work, we investigated the problem of computing low-entropy couplings for large support distributions, making three main contributions. First, we unified existing algorithms under the formalism of partition sets. Second, using this unified perspective, we introduced ARIMEC—the first general approach to computing low-entropy couplings for large-support distributions that can be applied to arbitrary distributions. Finally, we empirically showed the utility of ARIMEC in MCG and steganography applications. We commit to releasing our codebase as a documented package for computing low-entropy couplings for large-support distributions and hope others will find it useful.

For future work, there are a few application directions in which it would be interesting to push further with ARIMEC. First is unencrypted steganography. This direction is exciting because ARIMEC can achieve high throughput rates, as we observed in Figure 5, and because minimum-entropy coupling's usage for steganography was only recognized recently (Schroeder de Witt et al., 2023). Thus, there may be real-world settings in which it is applicable, especially since unencrypted steganography requires no key exchange. Second, because ARIMEC is the first IMEC algorithm capable of handling arbitrary discrete distributions, it opens the door to using large support distributions for classical minimum-entropy coupling applications in which the distributions may be non-factorable, such as entropic causal inference, random number generation, functional representations, and dimensionality reduction.

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

## A  INVERSE GENERATIVE PROCESSES

---
**Algorithm 4** TIMEC: $X \mid Y = y$

---
**procedure** IMEC($\mu, \nu, y$)
$\quad \gamma(X) \leftarrow \mu(X)$
$\quad$**for** $j = 1, \ldots, m$ **do**
$\quad\quad \gamma(X, Y_j \mid y_{1:j-1}) \leftarrow \text{MEC}(\gamma(X \mid y_{1:j-1}), \nu(Y_j \mid y_{1:j-1}))$
$\quad$**end for**
$\quad X \sim \gamma(X \mid y)$
$\quad$return $X$
**end procedure**

---

---
**Algorithm 5** FIMEC: $X \mid Y = y$

---
**procedure** IMEC($\mu, \nu, y$)
$\quad \gamma(X) \leftarrow \mu(X)$
$\quad$**for** $j = 1, \ldots, m$ **do**
$\quad\quad i^* \leftarrow \text{argmax}_i \mathcal{H}(\gamma(X_i \mid y_{1:j-1}))$
$\quad\quad \gamma(X_{i^*}, Y_j \mid y_{1:j-1}) \leftarrow \text{MEC}(\gamma(X_{i^*} \mid y_{1:j-1}), \nu(Y_j \mid y_{1:j-1}))$
$\quad\quad \gamma(X, Y_j \mid y_{1:j-1}) \leftarrow \gamma(X_{i^*}, Y_j \mid y_{1:j-1}) \cdot \left( \prod_{i \neq i^*} \gamma(X_i \mid y_{1:j-1}) \right)$
$\quad$**end for**
$\quad X \sim \gamma(X \mid y)$
$\quad$return $X$
**end procedure**

---

---
**Algorithm 6** IMEC (Generic Form): $X \mid Y = y$

---
**procedure** IMEC($\mu, \nu, y, \mathfrak{U}$)
$\quad \gamma(X) \leftarrow \gamma(\mu)$
$\quad$**for** $j = 1, \ldots, m$ **do**
$\quad\quad \mathcal{P} \leftarrow \arg\max_{\mathcal{P} \in \mathfrak{U}} \mathcal{H}(\gamma(B_{\mathcal{P}} \mid y_{1:j-1}))$
$\quad\quad \gamma(B_{\mathcal{P}}, Y_j \mid y_{1:j-1}) \leftarrow \text{MEC}(\gamma(B_{\mathcal{P}} \mid y_{1:j-1}), \nu(Y_j \mid y_{1:j-1}))$
$\quad$**end for**
$\quad X \sim \gamma(X \mid y)$
$\quad$return $X$
**end procedure**

---

## B  THEORY

### B.1  RUNTIME COMPLEXITY

**Proposition 2.1** (TIMEC Runtime). *Algorithm 1 can be implemented in $O(m \max(M, |\mathbb{X}|) \log \max(M, |\mathbb{X}|))$ time, where $M = \max_j |\mathbb{Y}_j|$.*

*Proof.* TIMEC runs for $m$ iterations. In each iteration, it performs an approximate minimum-entropy coupling, which costs $\max(M, |\mathbb{X}|) \log \max(M, |\mathbb{X}|)$ time. It also:

1. Performs a marginalization over $|\mathbb{X}|$ terms to compute $\gamma(Y_j \mid x, Y_{1:j-1})$.

2. Samples $Y_j$ from a distribution over $M$ terms.

3. Computes the conditional distribution $\gamma(X \mid Y_{1:j})$ over $|\mathbb{X}|$ terms.

However, these costs are dominated by $\max(M, |\mathbb{X}|) \log \max(M, |\mathbb{X}|)$. Thus the overall cost is $O(m \max(M, |\mathbb{X}|) \log \max(M, |\mathbb{X}|))$. $\qquad\square$

**Proposition 2.2** (FIMEC Runtime)**.** *Let Assumption 2.4 hold. Then Algorithm 2 can be implemented in $O(m \max(M, N) \log \max(M, N) + nN + m \log n + n \log n)$ time, where $N = \max_i |\mathbb{X}_i|$ and $M = \max_j |\mathbb{Y}_j|$.*

*Proof.* First, consider the upfront cost of computing the entropy of the components $X_1, \ldots, X_n$. There are $n$ components. For each component, computing the entropy can be achieved in $O(N)$ time. Thus, this requires $O(nN)$ time.

Next consider the upfront cost of sorting the components by entropy. Since there are $n$ blocks, this requires $O(n \log n)$ time.

Now consider the main loop of the algorithm. There are $m$ loops. In each loop, the block for which entropy was previously updated must be reinserted into the sorted list; this requires $O(\log n)$ time. Also, an approximate coupling is performed; this requires $O(\max(M, N) \log \max(M, N))$ time. It also

1. Performs a marginalization over $N$ terms to compute $\gamma(Y_j \mid x, Y_{1:j-1})$.

2. Samples $Y_j$ from a distribution over $M$ terms.

3. Computes the conditional distribution $\gamma(X \mid Y_{1:j})$ over $N$ terms.

However, these costs are dominated by the minimum-entropy coupling cost. Thus, the main loop requires $O(m \max(M, N) \log \max(M, N) + m \log n)$ time.

Therefore, the total runtime is $O(m \max(M, N) \log \max(M, N) + nN + m \log n + n \log n)$. $\quad\square$

**Proposition 4.1** (ARIMEC Runtime)**.** *Algorithm 3 with $\mathfrak{U}$ set to the prefix tree partition set runs in $O(m \max(M, N) \log \max(M, N) + mZN)$, where $N = \max_i |\mathbb{X}_i|$ and $M = \max_j |\mathbb{Y}_j|$ and where $Z$ is (a function of $n$ and $N$) that denotes the number of nodes in the prefix tree that require checking to find the maximum-entropy partition.*

*Proof.* ARIMEC runs for $m$ iterations. In each iteration, it computes the maximum-entropy partition. This requires checking $Z$ nodes, by definition. At each node checked, it must

1. Compute the updated posterior; by Proposition B.1, this costs $O(N)$.

2. Compute the associated entropy; this costs $O(N)$.

Thus we pay $O(ZN)$ for finding the maximum-entropy partition.

In each iteration, ARIMEC also computes an approximate minimum-entropy coupling; this requires $\max(M, N) \log \max(M, N)$ time. It also

1. Performs a marginalization over $N$ terms to compute $\gamma(Y_j \mid x, Y_{1:j-1})$.

2. Samples $Y_j$ from a distribution over $M$ terms.

3. Computes the conditional distribution $\gamma(X \mid Y_{1:j})$ over $N$ terms.

However, these costs are dominated by the minimum-entropy coupling cost.

Thus, we the total runtime is $O(m \max(M, N) \log \max(M, N) + mZN)$. $\quad\square$

### B.2   COUPLING

**Proposition 3.1** (Coupling)**.** *IMEC induces a coupling of $\mu$ and $\nu$.*

*Proof.* We proceed by induction on $m$. For the base case, consider $m = 1$. Then for any $y \in \mathbb{Y}$

$$\sum_{x \in \mathbb{X}} \gamma(x, y) = \sum_{x \in \mathbb{X}} \mu(x)\gamma(y \mid x) \tag{2}$$

$$= \sum_{\mathbb{B} \in \mathcal{P}^{(1)}} \sum_{x \in \mathbb{B}} \mu(x)\gamma(y \mid \mathbb{B}) \tag{3}$$

$$= \sum_{\mathbb{B} \in \mathcal{P}^{(1)}} \gamma(y \mid \mathbb{B}) \sum_{x \in \mathbb{B}} \mu(x) \tag{4}$$

$$= \sum_{\mathbb{B} \in \mathcal{P}^{(1)}} \gamma(y \mid \mathbb{B})\mu(\mathbb{B}) \tag{5}$$

$$= \sum_{\mathbb{B} \in \mathcal{P}^{(1)}} \gamma(y, \mathbb{B}) \tag{6}$$

$$= \nu(y), \tag{7}$$

where $\mathcal{P}^{(m)}$ denotes the partition selected at step $m$. Step (2) follows from chain rule; step (3) follows by construction; step (6) follows by chain rule; step (7) follows by the definition of a coupling. Now assume the result holds up to $m = \bar{m}$ and consider $m = \bar{m} + 1$. Observe, for any $y \in \mathbb{Y}$

$$\sum_{x \in \mathbb{X}} \gamma(x, y) = \sum_{x \in \mathbb{X}} \mu(x)\gamma(y_{1:\bar{m}} \mid x)\gamma(y_{\bar{m}+1} \mid x, y_{1:\bar{m}}) \tag{8}$$

$$= \sum_{\mathbb{B} \in \mathcal{P}^{(\bar{m}+1)}} \sum_{x \in \mathbb{B}} \gamma(y_{1:\bar{m}}, x)\gamma(y_{\bar{m}+1} \mid \mathbb{B}, y_{1:\bar{m}}) \tag{9}$$

$$= \sum_{\mathbb{B} \in \mathcal{P}^{(\bar{m}+1)}} \gamma(y_{\bar{m}+1} \mid \mathbb{B}, y_{1:\bar{m}}) \sum_{x \in \mathbb{B}} \gamma(y_{1:\bar{m}}, x) \tag{10}$$

$$= \sum_{\mathbb{B} \in \mathcal{P}^{(\bar{m}+1)}} \gamma(y_{\bar{m}+1} \mid \mathbb{B}, y_{1:\bar{m}})\gamma(y_{1:\bar{m}}, \mathbb{B}) \tag{11}$$

$$= \sum_{\mathbb{B} \in \mathcal{P}^{(\bar{m}+1)}} \gamma(y, \mathbb{B}, y_{1:\bar{m}}) \tag{12}$$

$$= \nu(y). \tag{13}$$

Step (8) follows from chain rule; step (9) follows by construction; step (12) follows by chain rule; step (13) follows by definition of a coupling. $\square$

### B.3 GREEDINESS

**Proposition 3.2** (Greediness). *If the trivial partition (i.e., the partition of singletons) is in $\mathfrak{U}$, IMEC approximately minimizes $\mathcal{H}(X, Y_{1:j})$ subject to $\mu, \nu, \gamma(X, Y_{1:j-1})$ on the $j$th iteration, for each $j$.*

*Proof.* Consider that performing a coupling with the trivial partition (or a partition that it is equivalent up to elements with zero probability) is equivalent to performing a partition with $\mathbb{X}$ itself. Then, invoking Lemma B.1, it suffices to show that the statement holds for $\mathbb{X}$.

To see this, first recall

$$\mathcal{H}(X, Y) = \mathcal{H}(Y \mid X) + \mathcal{H}(X)$$

Because the entropy of $X$ is fixed (as it is determined by its marginal $\mu$), minimum-entropy coupling is equivalent to minimum-conditional-entropy coupling. Then, note that, by chain rule, we have

$$\mathcal{H}(Y_{1:j} \mid X) = \sum_{k=1}^{j} \mathcal{H}(Y_k \mid X, Y_{1:k-1}) = \mathcal{H}(Y_j \mid X, Y_{1:j-1}) + \sum_{k=1}^{j-1} \mathcal{H}(Y_k \mid X, Y_{1:k-1}).$$

At iteration $j$, all terms below $j$ have already been determined. Thus, the rightmost summation term is fixed and minimizing $\mathcal{H}(X, Y_{j-1})$ is reduced to minimizing $\mathcal{H}(Y_j \mid X, Y_{1:j-1})$. By again invoking the equivalence between minimum-entropy coupling and minimum-conditional-entropy coupling, this is equivalent to minimizing $\mathcal{H}(X, Y_j \mid Y_{1:j-1})$, which is exactly what IMEC minimizes at iteration $j$. $\square$

**Lemma B.1.** *Let $\mathfrak{U}$ be the set of all partitions over $\mathbb{X}$. For any distribution over $\mathbb{X}$, any maximum-entropy partition is equivalent to the trivial partition (i.e., the partition of singletons) up to zero-probability elements.*

*Proof.* Consider a block $\mathbb{B}$ of some partition $\mathcal{P}$ of $\mathbb{X}$. The entropy that $\mathbb{B}$ contributes is

$$-\gamma(\mathbb{B})\log\gamma(\mathbb{B}).$$

The first derivative of this function is

$$-\log\gamma(\mathbb{B})-1.$$

The second derivative is

$$-\frac{1}{\gamma(\mathbb{B})}.$$

Since the second derivative is always negative, the contribution of $\mathbb{B}$ to the total entropy is strictly concave. Thus, further subdividing $\mathbb{B}$ increases its contribution to total entropy, up to elements with zero probability. $\qquad\square$

## B.5 POSTERIOR UPDATES

**Proposition B.1** (Posterior Updates). *Let $(\mathbb{V}, \mathbb{E})$ be the prefix tree for $\mathbb{X}$. Assume that the posterior over a partition is updated if and only if its corresponding node is touched and that nodes are touched by traversing edges of the tree. Let $\mathcal{P}_v$ be a partition whose posterior was updated on the current iteration $j$. If $(v, c) \in \mathbb{E}$ and $c$ was last visited on iteration $j' \leq j$, then*

$$\gamma(\mathbb{B}_{c\not\sqsubset} \mid Y_{1:j}) = 1 - \gamma(\mathbb{B}_{c\sqsubset} \mid y_{1:j})$$

*and, for $\mathbb{B}' \in \mathcal{P}_c$ where $\mathbb{B}' \neq \mathbb{B}_{c\not\sqsubset}$,*

$$\gamma(\mathbb{B}' \mid y_{1:j}) \propto \gamma(\mathbb{B}' \mid y_{1:j'}).$$

*If $(p, v) \in \mathbb{E}$ and $p$ was last visited on iteration $j' \leq j$, then*

$$\gamma(\mathbb{B}_{v\sqsubset} \mid Y_{1:j}) = 1 - \gamma(\mathbb{B}_{v\not\sqsubset} \mid y_{1:j})$$

*and, for $\mathbb{B}' \in \mathcal{P}_p$ where $\mathbb{B}' \neq \mathbb{B}_{v\sqsubset}$,*

$$\gamma(\mathbb{B}' \mid y_{1:j}) \propto \gamma(\mathbb{B}' \mid y_{1:j'}).$$

*Proof.* First consider that $(\mathbb{B}_{c\not\sqsubset}, \mathbb{B}_{c\sqsubset})$ and $(\mathbb{B}_{v\sqsubset}, \mathbb{B}_{v\not\sqsubset})$ are pairs of complementary events. Thus, their probabilities must sum to one by the complement rule.

Now, consider that, since $v$ is not an extension of $c$, if $c$ was last visited on iteration $j'$, it follows that no extension of $c$ can have been visited since iteration $j'$. (This follows because every path from a extension of $c$ to $v$ must touch $c$.) Therefore, every partition updated since $\mathcal{P}_c$ was last updated must correspond to a vertex that is not an extension of $c$. Thus, posterior updates can only change the probability of blocks of $\mathcal{P}_c$ corresponding to extensions of $c$ by a constant factor.

Similarly, consider that, since $v$ is a extension of $p$, if $p$ was last visited on iteration $j'$, it follows all vertices that have been visited since iteration $j'$ are extensions of $v$. (This follows because every path from $v$ to a node that is not a extension of $v$ must touch $p$.) Therefore, every partition used for coupling onward from iteration $j'$ corresponds to a vertex that is an extension of $v$. Thus, posterior updates can only change the probability of blocks of $\mathcal{P}_p$ not corresponding to extensions of $v$ by a constant factor. $\qquad\square$

## B.6 ENTROPY UPPER BOUND

**Lemma B.2** (Entropy Upper Bound). *Let $\mu$ be a probability distribution over $\kappa$ elements. Fix any element $\mu(x^*)$. Then for any $q$ such that $\frac{1}{\kappa} \leq q \leq \mu(x^*)$, we have*

$$\mathcal{H}(\mu) \leq \begin{cases} -q\log q - (1-q)\log\frac{(1-q)}{\kappa-1} & q \in [1/\kappa, 1) \\ 0 & q = 1. \end{cases}$$

*Proof.* First note that if $\mu(x^*) = 1$ then $\mathcal{H}(\mu) = 0$ and the upper bound holds trivially.

Next, consider the case in which $\mu(x^*) < 1$. We will show that this upper bound holds in the case when $q = \mu(x^*)$. We first observe that the entropy is given by

$$\mathcal{H}(\mu) = -\sum_x \mu(x) \log \mu(x) = -\mu(x^*) \log \mu(x^*) - \sum_{x \neq x^*} \mu(x) \log \mu(x)$$

Now, we can consider another probability distribution $\mu'$ over $n - 1$ values (everything except $x^*$), which is given by $\mu'(x) = \frac{\mu(x)}{1-\mu(x^*)}, \forall x \neq x^*$. Since entropy is maximized by a uniform distribution, we have that $\mathcal{H}(\mu') \leq -\log(\frac{1}{n-1})$.

We observe that

$$\mathcal{H}(\mu') = -\sum_{x \neq x^*} \mu'(x) \log \mu'(x)$$

$$= -\frac{1}{(1 - \mu(x^*))} \sum_{x \neq x^*} \mu(x) \log \mu'(x)$$

$$= -\frac{1}{(1 - \mu(x^*))} \sum_{x \neq x^*} \mu(x) \Big( \log \mu(x) - \log(1 - \mu(x^*)) \Big)$$

$$= -\frac{1}{(1 - \mu(x^*))} \left[ \sum_{x \neq x^*} \Big( \mu(x) \log \mu(x) \Big) + \log(1 - \mu(x^*)) \right]$$

Then, plugging this into the inequality for $\mathcal{H}(\mu')$ gives us that

$$-\sum_{x \neq x^*} \mu(x) \log \mu(x) \leq (1 - \mu(x^*)) \left( -\log\left(\frac{1}{n-1}\right) - \log(1 - \mu(x^*)) \right)$$

$$= -(1 - \mu(x^*)) \log\left(\frac{1 - \mu(x^*)}{n - 1}\right)$$

Thus, this gives us that

$$\mathcal{H}(\mu) \leq -\mu(x^*) \log \mu(x^*) - (1 - \mu(x^*)) \log\left(\frac{1 - \mu(x^*)}{n - 1}\right)$$

as desired.

Next, we will show that this upper bound decreases in $q$. We can consider taking the partial derivative with the upper bound with respect to $q$, which gives us that

$$D_q \left( -q \log q - (1 - q) \log \frac{(1 - q)}{n - 1} \right) = -\log q - 1 + \log \frac{(1 - q)}{n - 1} + 1 = -\log q + \log \frac{(1 - q)}{n - 1}.$$

Setting this equal to zero gives us that

$$\log q - \log \frac{1 - q}{n - 1} = 0$$

$$\implies q = \frac{1}{n}.$$

Next, we observe that the second derivative of the upper bound with respect to $q$ is given by

$$D_q D_q \left( -q \log q - (1 - q) \log \frac{(1 - q)}{n - 1} \right) = D_q \left( -\log q + \log \frac{(1 - q)}{n - 1} \right) = \frac{1}{q(q - 1)}.$$

Thus, this is negative for all values of $0 < q < 1$, which gives us that the upper bound is decreasing in $q$ on the interval $[\frac{1}{n}, 1)$. Therefore, since it holds for $q = \mu(x^*)$, it must hold for $q \in [1/n, \mu(x^*)]$. $\square$

**Proposition B.2** (Maximum-Entropy Partition). *Let $\kappa$ be an upper bound on the number of blocks with positive probability in the prefix tree partition set:*

$$\kappa = \max_{\mathcal{P} \in \mathfrak{U}} |\{\mathbb{B} \in \mathcal{P} \mid \mu(\mathbb{B}) > 0\}|.$$

*Denote*

$$\mathcal{U} \colon q \mapsto -q \log q - (1 - q) \log \frac{1 - q}{\kappa - 1}.$$

*For any $c$ such that $(v, c) \in \mathbb{E}$, if*

$$\gamma(\mathbb{B}_{c\sqsubset} \mid y_{1:j}) < 1 - 1/\kappa,$$

*then, for all $u$ such that $c \sqsubset u$,*

$$\mathcal{H}(\gamma(B_{\mathcal{P}_u} \mid y_{1:j})) \leq \mathcal{U}(\gamma(\mathbb{B}_{c\not\sqsubset} \mid y_{1:j})).$$

*Also, if*

$$\gamma(\mathbb{B}_{v\not\sqsubset} \mid y_{1:j}) < 1 - 1/\kappa$$

*then, for all $u$ such that $v \not\sqsubset u$,*

$$\mathcal{H}(\gamma(B_{\mathcal{P}_u} \mid y_{1:j})) \leq \mathcal{U}(\gamma(\mathbb{B}_{v\sqsubset} \mid y_{1:j})).$$

*Proof.* Observe

$$
\begin{aligned}
& \gamma(\mathbb{B}_{c\sqsubset} \mid y_{1:j}) < 1 - 1/\kappa \\
\iff & -\gamma(\mathbb{B}_{c\sqsubset} \mid y_{1:j}) > -1 + 1/\kappa \\
\iff & 1 - \gamma(\mathbb{B}_{c\sqsubset} \mid y_{1:j}) > 1/\kappa \\
\iff & \gamma(\mathbb{B}_{c\not\sqsubset} \mid y_{1:j}) > 1/\kappa,
\end{aligned}
$$

where the last line holds because $(\mathbb{B}_{c\not\sqsubset}, \mathbb{B}_{c\sqsubset})$ are complementary events. Then note, at any $u$ such that $c \sqsubset u$, there must exist a block $\mathbb{B}_{u\not\sqsubset} \in \mathcal{P}_u$ such that $\mathbb{B}_{c\not\sqsubset} \subset \mathbb{B}_{u\not\sqsubset}$. Therefore, we have $\gamma(\mathbb{B}_{u\not\sqsubset} \mid y_{1:j}) \geq \gamma(\mathbb{B}_{c\not\sqsubset} \mid y_{1:j})$. The bound follows from applying Lemma B.2.

Similarly, observe

$$
\begin{aligned}
& \gamma(\mathbb{B}_{v\not\sqsubset} \mid y_{1:j}) < 1 - 1/\kappa \\
\iff & -\gamma(\mathbb{B}_{v\not\sqsubset} \mid y_{1:j}) > -1 + 1/\kappa \\
\iff & 1 - \gamma(\mathbb{B}_{v\not\sqsubset} \mid y_{1:j}) > 1/\kappa \\
\iff & \gamma(\mathbb{B}_{v\sqsubset} \mid y_{1:j}) > 1/\kappa.
\end{aligned}
$$

Select any $u$ such that $v \not\sqsubset u$. If $u \sqsubset v$, then there exists $u'$ such that $(u, u') \in \mathbb{E}$ and $u' \sqsubset v$. Furthermore, $\mathbb{B}_{u'\sqsubset} \in \mathcal{P}_u$ and $\mathbb{B}_{v\sqsubset} \subset \mathbb{B}_{u'\sqsubset}$. Therefore, we have $\gamma(\mathbb{B}_{u'\sqsubset} \mid y_{1:j}) \geq \gamma(\mathbb{B}_{v\sqsubset} \mid y_{1:j})$.

On the the other hand, if $u \not\sqsubset v$, then $\mathbb{B}_{v\sqsubset} \subset \mathbb{B}_{u\not\sqsubset}$. Thus $\gamma(\mathbb{B}_{u\not\sqsubset} \mid y_{1:j}) \geq \gamma(\mathbb{B}_{v\sqsubset} \mid y_{1:j})$.

Thus, $\mathcal{P}_u$ possesses a block with probability at least as great as that of $\mathbb{B}_{v\sqsubset}$. The bound follows from the application of Lemma B.2. $\qquad\square$

## C  VISUALIZATIONS

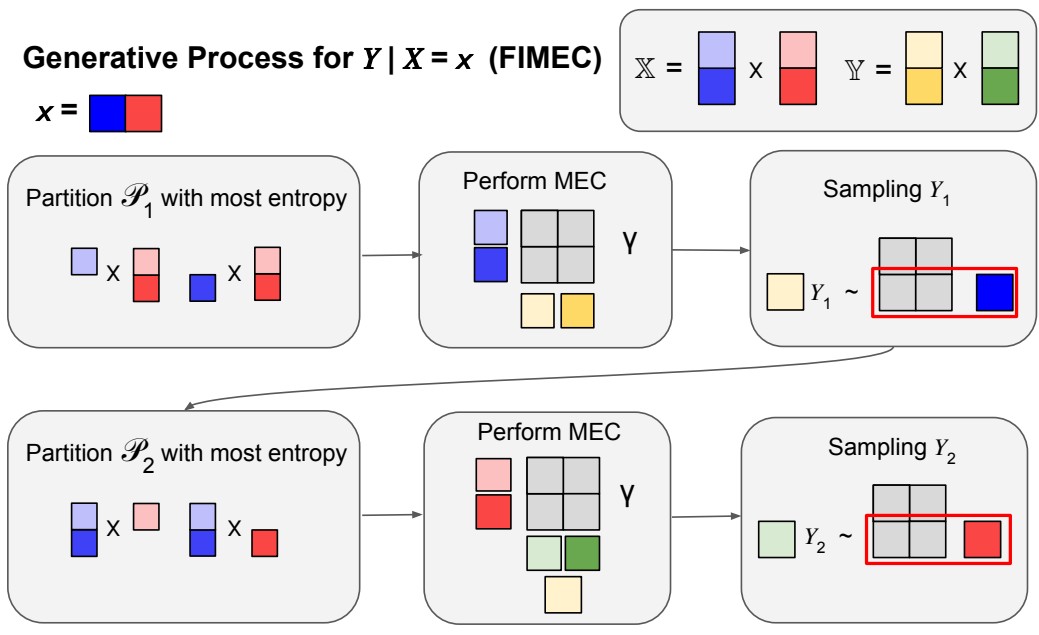

Figure 6: Visualization of two iterations of FIMEC.

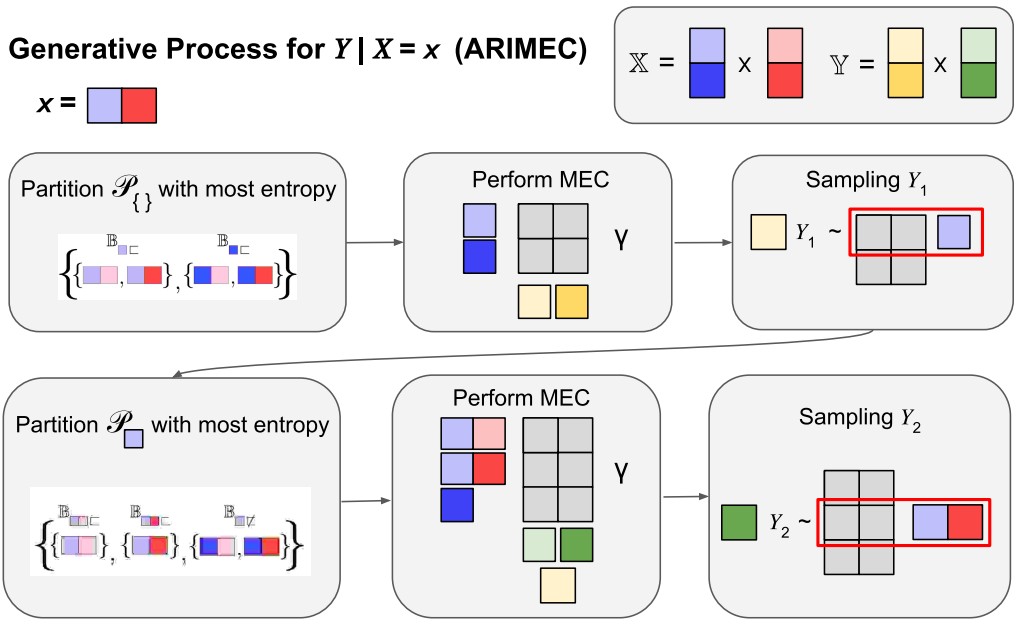

Figure 7: Visualization of two iterations of ARIMEC.

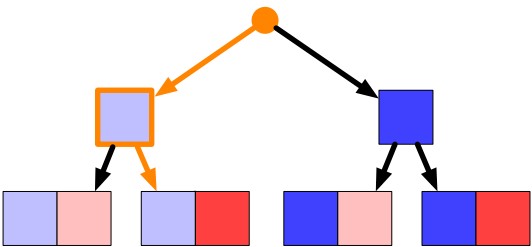

Figure 8: Path shown down the prefix tree corresponding to the procedure in Figure 7.

As ARIMEC builds certainty in the value of $X$, it progresses down the tree; thus, ARIMEC can be visualized as iteratively traversing down (and sometimes backtracking up) the tree structure. This process is illustrated in Figure 7, and the corresponding path down the tree is shown in Figure 8.

## D    EXPERIMENTS

### D.1    MARKOV CODING GAMES

Sokota et al. (2022) specify Markov coding games as the following setting:

> An MCG is a tuple $\langle (\mathcal{S}, \mathcal{A}, \mathcal{T}, \mathcal{R}), \mathcal{M}, \mu, \zeta \rangle$, where $(\mathcal{S}, \mathcal{A}, \mathcal{T}, \mathcal{R})$ is a Markov decision process, $\mathcal{M}$ is a set of messages, $\mu$ is a distribution over $\mathcal{M}$ (i.e., the prior over messages), and $\zeta$ is a non-negative real number we call the message priority.
> **An MCG proceeds in the following steps:**
> 1. First, a message $M \sim \mu$ is sampled from the prior over messages and revealed to the sender.
> 2. Second, the sender uses a message conditional policy $\pi_{|M}$, which takes states $s \in \mathcal{S}$ and messages $m \in \mathcal{M}$ as input and outputs distributions over MDP actions $\Delta(\mathcal{A})$, to generate a trajectory $Z \sim (\mathcal{T}, \pi_{|M})$ from the MDP.
> 3. Third, the sender's terminal MDP trajectory $Z$ is given to the receiver as an observation.
> 4. Fourth, the receiver uses a terminal MDP trajectory conditional policy $\pi_{|Z}$, which takes terminal trajectories $z \in \mathcal{Z}$ as input and outputs distributions over messages $\Delta(\mathcal{M})$, to estimate the message $\hat{M} \sim \pi_{|Z}(Z)$.
>
> The objective of the agents is to maximize the expected weighted sum of the return and the accuracy of the receiver's estimate $\mathbb{E}\left[ \mathcal{R}(Z) + \zeta \mathbb{I}[M = \hat{M}] \mid \pi_{|M}, \pi_{|Z} \right]$. Optionally, in cases in which a reasonable distance function is available, we allow for the objective to be modified to minimizing the distance between the message and the guess $d(M, \hat{M})$, rather than maximizing the probability that the guess is correct.

### D.2    MEME

Sokota et al. (2022) specify MEME as follows:

> **Step One: Maximum Entropy Reinforcement Learning** In the first step, MEME uses MaxEnt RL to construct an MDP policy $\pi$. This policy is an MDP policy, not an MCG policy, and therefore does not depend on the message. Note that this policy depends on the choice of temperature $\alpha$ used for the MaxEnt RL algorithm.
> **Step Two: Minimum Entropy Coupling** In the second step, at execution time, MEME constructs a message-conditional policy online using MECs. Say that, up to time $t$, the sender is in state $s^t$, history $h^t$ and has played according to the state and message conditional policy $\pi_{|M}^{:t}$ so far. Let

$$b^t = \mathcal{P}(M \mid h^t, \pi_{|M}^{:t})$$

be the posterior over the message, conditioned on the history and the historical policy. MEME performs a MEC between the posterior over the message $b^t$ and distribution over actions $\pi(s^t)$, as determined by the MDP policy. Let $\nu = \text{MEC}(b^t, \pi(s^t))$ denote joint distribution over messages and actions resulting from the coupling. Then MEME sets the sender to act according to the message conditional distribution

$$\pi^t_{|M}(s^t, m) = \nu(A^t \mid M = m)$$

of the coupling distribution $\nu = \text{MEC}(b^t, \pi(s^t))$.

Given the sender's MDP trajectory, MEME's receiver uses the sender's MDP policy and MEC procedure to reconstruct the sender's message conditional policy along the trajectory; thereafter, the receiver computes the posterior and guesses the maximum a posteriori (MAP) message.

### D.3  INFORMATION-THEORETIC STEGANOGRAPHY

Schroeder de Witt et al. (2023) summarize Cachin (1998)'s information-theoretic steganography setting as follows:

> **Problem Setting** The objects involved in information-theoretic steganography can be divided into two classes: those which are externally specified and those which require algorithmic specification. Each class contains three objects. The externally specified objects include the distribution over plaintext messages $\mathcal{M}$, the distribution over covertext $\mathcal{C}$, and the random source generator.
>
> - The distribution over plaintext messages $\mathcal{M}$ may be known by the adversary, but is not known by the sender or the receiver. However, the sender and receiver are aware of the domain $\mathbb{M}$ over which $\mathcal{M}$ ranges. The sampled plaintext message $M$ is explicitly known by the sender, but not to the receiver or the adversary.
> - The covertext distribution $\mathcal{C}$ is assumed to be known by the sender, the receiver, and the adversary.
> - The random source generator provides the sender with a mechanism to take random samples from distributions. This random source is known to the sender but not to the receiver or adversary.
>
> The objects requiring algorithmic specification, which are collectively referred to as a stegosystem, are the key generator, the encoder, and the decoder.
>
> - The key generator produces a private key $K$ in the form of a binary string. This private key is shared between the sender and receiver over a secure channel prior to the start of the stegoprocess and can be used to coordinate encryption and decryption. The key generation process may be known to the adversary, but the realization of the key $K$ is not.
> - The encoder takes a private key $K$, a plaintext message $M$, and a source of randomness $R$ as input and produces a stegotext $S$ in the space of covertexts $\mathbb{C}$.
> - The decoder takes a private key $K$ and a stegotext $S$ as input and returns an estimated plaintext message $\hat{M}$.

They specify the following objectives and methodological outline for the setting:

> **Definition D.1.** *(Cachin, 1998) Given covertext distribution $\mathcal{C}$ and plaintext message space $\mathbb{M}$, a stegosystem is $\epsilon$-secure against passive adversaries if the KL divergence between the distribution of covertext $\mathcal{C}$ and the distribution of stegotext $\mathcal{S}$ less than $\epsilon$; i.e., $KL(\mathcal{C}, \mathcal{S}) < \epsilon$. It is perfectly secure if the KL divergence is zero; i.e., $KL(\mathcal{C}, \mathcal{S}) = 0$.*
>
> In other words, a steganographic system is perfectly secure if the distribution of stegotext $\mathcal{S}$ communicated by the sender is exactly the same as the distribution of covertext $\mathcal{C}$.
>
> In addition to security, it is desirable for stegosystems to transmit information efficiently. Mutual information between messages and stegotexts is one way to quantify efficiency.

**Definition D.2.** *The mutual information $\mathcal{I}(M;S) = \mathcal{H}(M) - \mathcal{H}(M \mid S)$ between the message $M$ and stegotext $S$ is the expected amount of uncertainty in the message $M$ that is removed by conditioning on the stegotext $S$.*

**Methodological Outline** A common class of stegosystems uses two-step encoding and two-step decoding processes, as described below:

1. The sender uses the private key $K$ to injectively map the plaintext message $M$ into ciphertext $\mathbb{X} = \{0,1\}^\ell$ in such a way that the induced distribution over ciphertext $\mathcal{X}$ is uniformly random, regardless of the distribution of $\mathcal{M}$.[5]

2. The sender uses a (potentially stochastic) mapping $f \colon \mathbb{X} \rightsquigarrow \mathbb{C}$ to transform the ciphertext $X$ into stegotext $S$ (which exists in the space of covertexts $\mathbb{C}$).

3. The receiver estimates the ciphertext $\hat{X}$ from the stegotext $S$.

4. The receiver inverts the estimated ciphertext $\hat{X}$ to a plaintext message $\hat{M}$ with private key $K$.[6]

Given the definition below Schroeder de Witt et al. (2023) show the following guarantees:

**Definition D.3.** *We say that an encoding procedure $f \colon \mathbb{X} \rightsquigarrow \mathbb{C}$ is induced by a coupling if there exists $\gamma \in \Gamma(\mathcal{X}, \mathcal{C})$ such that for all $x \in \mathbb{X}, c \in \mathbb{C}, \mathcal{P}(f(x){=}c) = \gamma(C{=}c \mid X{=}x)$.*

**Theorem D.4.** *A steganographic encoding procedure is perfectly secure if and only if it is induced by a coupling.*

**Theorem D.5.** *Among perfectly secure encoding procedures, a procedure $f \colon \mathbb{X} \rightsquigarrow \mathbb{C}$ maximizes the mutual information $\mathcal{I}(M;S)$ if and only if $f$ is induced by a minimum entropy coupling.*

## D.4 UNENCRYPTED STEGANOGRAPHY

In the unencrypted steganography setting we consider, there are similarly two classes of objects: those which are externally specified and those which require algorithmic specification. The externally specified objects include the distribution over plaintext messages $\mathcal{M}$, the distribution over covertext $\mathcal{C}$, and the random source generator. The assumptions regarding the covertext and random source generator work the same way as in Cachin (1998)'s setting. In contrast, the assumptions regarding the distribution over plaintext messages $\mathcal{M}$ differ significantly: Specifically, we assume that $\mathcal{M}$ is known to both the sender and receiver but that it is *not known* by the adversary. Furthermore, we require that messages are sampled independently from this distribution (which is not required under Cachin (1998)'s setup. The objects requiring algorithmic specification are the encoder and decoder. These work the same way as in information-theoretic steganography, except that they do not take a private key as input.

Similarly to information-theoretic steganography, one objective of unencrypted steganography setting we consider is to maximize the mutual information $\mathcal{I}(M;S)$ between the plaintext message $M$ and the stegotext $S$. However, instead of prioritizing security as defined in Definition D.1, the unencrypted setting prioritizes undetectability:

**Definition D.6.** *Given a covertext distribution $\mathcal{C}$ and a plaintext message distribution $\mathcal{M}$, a stegosystem is $\epsilon$-undetectable against passive adversaries if the KL divergence between the distribution of covertext $\mathcal{C}$ and the distribution of stegotext $\mathcal{S}$ is less than $\epsilon$; i.e., $KL(\mathcal{C}, \mathcal{S}) < \epsilon$. It is perfectly undetectable if the KL divergence is zero; i.e., $KL(\mathcal{C}, \mathcal{S}) = 0$.*

For this setting, we consider encoding procedures of the form $f \colon \mathbb{M} \rightsquigarrow \mathbb{C}$. Then, using the same analogous proofs as Schroeder de Witt et al. (2023), the following results are immediate.

---

[5]For example, if $K$ is drawn from a uniform random distribution, $\text{bin}(M)$ denotes a deterministic binarization of $M$, and XOR represents the component-wise exclusive-or function, then $X = \text{XOR}(\text{bin}(M), K)$ is guaranteed to be distributed uniformly randomly, regardless of the distribution of messages $\mathcal{M}$.

[6]For the example in footnote 5, the receiver can recover the binarized message $\text{bin}(M)$ using the mapping $X \mapsto \text{XOR}(X, K)$ and invert the binarization to recover the plaintext $M$.

**Theorem D.7.** *A steganographic encoding procedure* $f\colon \mathbb{M} \rightsquigarrow \mathbb{C}$ *is perfectly undetectable if and only if it is induced by a coupling.*

**Theorem D.8.** *Among perfectly secure encoding procedures, a procedure* $f\colon \mathbb{M} \rightsquigarrow \mathbb{C}$ *maximizes the mutual information* $\mathcal{I}(M; S)$ *if and only if* $f$ *is induced by a minimum-entropy coupling.*

Thus, we see that minimum-entropy coupling-based approaches are also well suited to unencrypted steganography.

**Discussion on Setting Assumptions**    The unencrypted steganography setting we consider has both significant advantages and disadvantages compared to Cachin (1998)'s setting. Disadvantages include:

1. Strong assumptions on the messages distribution: Both the assumption that the message distribution is known and that it produces messages independently across time are atypical and generally do not hold in practice.

2. Violation of Kerckhoff's principle: Kerckhoff's principle states that, even if an adversary has complete knowledge of the system aside from the private key, it should not compromise the security of the system. In the setting we consider, there is no private key; instead, the sender and receiver rely on "security through obscurity". Thus, an adversary with complete knowledge of the system would have the power to decode plaintext messages.

The advantages include:

1. Private key exchange is not required: The burden of private key exchange required in Cachin (1998)'s formulation is quite substantial. Specifically, the sender and receiver need to agree upon a fresh randomly generated key over a secure channel prior to every message exchange. In some settings, the existence of such a secure channel may void the need for steganography in the first place. In others, it may severely limit the frequency of communication due the expense of using such channels.

2. Higher potential information throughput: To achieve security guarantees, approaches to Cachin (1998)'s setting embed plaintext messages into randomized ciphertext. This randomized embedding increases the amount of information that needs to be encoded into the stegotext. Thus, avoiding this embedding step could yield efficiency guarantees. (And, indeed, we observe this to be the case in Figure 5.)

