# OpenReview forum: "Computing Low-Entropy Couplings for Large-Support Distributions"
_ICLR.cc/2024/Conference — Submitted to ICLR 2024_

### Official Review · Reviewer_n51E · 2023-10-29

**Soundness:** 3 good
**Presentation:** 4 excellent
**Contribution:** 3 good
**Rating:** 5
**Confidence:** 2

**Summary:**

The paper addresses the challenge of computing minimum-entropy couplings (MEC) for large-support distributions.
MEC aims to find a joint distribution with the least joint entropy given two specific marginal distributions.
Existing algorithms that find provable approximations for such couplings are unsuitable for very large-support distributions.
Current heuristic methods, called Iterative Minimum-Entropy Coupling (IMEC), have limitations in handling these distributions.

Contributions:
* Unified IMEC Algorithms: The authors provide a unified framework for IMEC algorithms using sets of partitions.
* ARIMEC Introduction: Leveraging the unified view, a new method, ARIMEC, is introduced. It computes low-entropy couplings for any large-support distribution. Efficiency improvements, like lazy updates and entropy bounds, are incorporated.
* Empirical Results: ARIMEC's effectiveness is showcased in Markov coding games and steganography, resulting in improved communication rates.

The paper presents a novel approach, ARIMEC, to compute low-entropy couplings for large-support distributions and validates its utility with real-world applications.
The authors promise to share their codebase with the community.

**Strengths:**

As the author states, this paper has three contributions. Let's discuss the strengths of each in order.

1. Unified IMEC Algorithms
The key strength of the unified IMEC framework, based on the description, seems to lie in its flexibility through the use of partitions.
Depending on how these partitions are selected, different algorithms or strategies can be realized.

2. ARIMEC Introduction
The proposed ARIMEC algorithm falls into the unified IMEC framework.
In the framework, irrespective of the specific autoregressive structure or the particularities of the marginal \mu,
the algorithm can always satisfy the three conditions: Condition 3.1, 3.2 and 3.3.
This capability is significant because it allows ARIMEC to be applied to a wide range of problems or datasets
that have an autoregressive nature without the need for tweaking the algorithm for each specific case.

3. Empirical Results
ilding on the work of Sokota et al. (2022), the authors explore an application of ARIMEC in the communication of messages
through Markov decision processes. It's a unique application that hasn't been frequently touched upon in literature.

**Weaknesses:**

* The experiments seem to be heavily centered around specific domains like Markov coding games and steganography.
  While these are valuable explorations, they might not give a complete picture of ARIMEC's versatility across other potential domains or applications.
  I'd like a bit more discussion on the potential applications of ARIMEC.

**Questions:**

* In Section 3, a unified view of IMEC is proposed, but are there any existing algorithms to be unified other than TIMEC and FIMEC?
* Figure 3, why do you compare Token-wise Error Rate?

---

> ### Author Response · Authors · 2023-11-17
> **Response to Reviewer n51E**
>
> Thanks for the insightful feedback. It has helped us improve the submission, and we've submitted an updated version.
>
> > The experiments seem to be heavily centered around specific domains like Markov coding games and steganography. While these are valuable explorations, they might not give a complete picture of ARIMEC's versatility across other potential domains or applications. I'd like a bit more discussion on the potential applications of ARIMEC.
>
>
> We agree that such a discussion is warranted. To address this concern, we added the following paragraph about additional potential applications for ARIMEC to the conclusion:
>
> "For future work, there are a few application directions in which it would be interesting to push further with ARIMEC. First is unencrypted steganography. This direction is exciting because ARIMEC can achieve high throughput rates, as we observed in Figure 5, and because minimum-entropy coupling's usage for steganography was only recognized recently Schroeder de Witt et al. (2023). Thus, there may be real-world settings in which it is applicable, especially since unencrypted steganography requires no key exchange. Second, because ARIMEC is the first IMEC algorithm capable of handling arbitrary discrete distributions, it opens the door to using large support distributions for classical minimum-entropy coupling applications in which the distributions may be non-factorable, such as entropic causal inference, random number generation, functional representations, and dimensionality reduction."
>
> > In Section 3, a unified view of IMEC is proposed, but are there any existing algorithms to be unified other than TIMEC and FIMEC?
>
> No, other than ARIMEC later in the submission.
>
> > Figure 3, why do you compare Token-wise Error Rate?
>
> We are trying to measure the error rate at which the language model text is decoded properly, as this is the metric of interest in the steganography use case of minimum-entropy coupling. This error rate is related to joint entropy, but not exactly the same, so it requires separate treatment.
>
> ---
>
> We thank the reviewer again for their feedback. Please let us know if you have any further suggestions for the revised text.

---

### Official Review · Reviewer_dCYg · 2023-10-31

**Soundness:** 3 good
**Presentation:** 3 good
**Contribution:** 3 good
**Rating:** 6
**Confidence:** 3

**Summary:**

The paper provides an efficient algorithm for min-entropy coupling that the authors call ARIMEC. The best previously known algorithm required one of the distributions to be  factorable into blocks with small supports, while their algorithm doesn't require that.

**Strengths:**

The result is new and seems to be strong. It answers an open question from Sokota et al. (2022). The paper is easy to read, the main result is clear and its comparison with the state of the art is explained. The idea is nice and simple.

**Weaknesses:**

Even though I like the idea, the approach is not very sophisticated, it combines some standard (but non-trivial) combinatorial techniques with the results of prior works.

Also, even though the paper is easy to read, there are no rigorous formulations of the results in the main part of the paper, and some statements are not very precise (e.g. I find usage of terms like "small" in formal conditions not very nice).

**Questions:**

I do not have any specific questions, since the paper basically solves the problem as it was stated in Sokota et al. (2022). Maybe just a high-level question: Did you think of any potential future directions where your approach could be useful?

Suggestions: As I said before, I would also write conditions in more formal manner (e.g. I recommend to replace "small" by something concrete and formal, and then add a high-level explanation below or above the formal definition). I also recommend to write your results as theorems (and keep current high-level explanations close to the formal statements).

---

> ### Author Response · Authors · 2023-11-17
> **Response to Reviewer dCYg**
>
> Thanks to the reviewer for the helpful review and comments. The feedback really improved our work, and we've uploaded a new version.
>
> > there are no rigorous formulations of the results in the main part of the paper, and some statements are not very precise (e.g. I find usage of terms like "small" in formal conditions not very nice) ... Suggestions: As I said before, I would also write conditions in more formal manner (e.g. I recommend to replace "small" by something concrete and formal, and then add a high-level explanation below or above the formal definition). I also recommend to write your results as theorems (and keep current high-level explanations close to the formal statements).
>
> We completely agree with the reviewer's criticism of the submission. To help address this, we removed the informal conditions and added runtime complexity statements for each of the algorithms (and their proofs in the Appendix).
>
> > Maybe just a high-level question: Did you think of any potential future directions where your approach could be useful?
>
> We are excited about further exploring of further developing unencrypted steganography, as its literature is relatively less developed and minimum-entropy coupling may facilitate high information throughput. We are also interested in looking into applications of ARIMEC in entropic causal inference, random number generation, functional representations, and dimensionality reduction, as these are all areas where approximate minimum-entropy coupling algorithms have proved useful. We added a discussion on this to the end of the conclusion in the revised version of the submission.
>
> ---
>
> We thank the reviewer again for their feedback. Please let us know if you have any further suggestions for the revised text.

---

> > ### Comment · Reviewer_dCYg · 2023-11-23
> >
> > Dear Authors,
> >
> > Thank you very much for your response! The score remains unchanged.

---

### Official Review · Reviewer_caMu · 2023-11-07

**Soundness:** 2 fair
**Presentation:** 2 fair
**Contribution:** 2 fair
**Rating:** 3
**Confidence:** 3

**Summary:**

Minimum entropy coupling (MEC) is the following problem. Given marginal distributions of two finitely supported random variables X and Y, find a joint distribution \gamma on X and Y (called a “coupling” of X and Y) s.t. the entropy of \gamma is as small as possible. In addition to being a natural problem from a theoretical point of view, MEC also has various practical applications, including steganography (the art of hiding secret messages in plain sight).

The authors propose a unified framework that captures previous approaches to MEC and propose a novel *heuristic* algorithm for MEC called ARIMEC. That is, their method does not have provable guarantees regarding the entropy achieved by the output coupling. The reason for this is that MEC is NP-hard (where the instance size is defined to be the support size of X and Y) and provable approximation algorithms for MEC run in O(N log N) time, where N is the support size. In many interesting practical applications, the support size N is often extremely large so even linear-in-N time is considered too slow. Thus, the authors turn to heuristic algorithms and argue for the utility of their approach via empirical evaluation.

**Strengths:**

The unifying framework using collections of partitions is relatively simple and nicely captures previous approaches, though its exposition could be improved. In addition, the application to steganography is interesting. Heuristic MEC algorithms, despite their lack of rigorous guarantees, could lead to interesting future work on steganography using large language models (LLMs) since for LLMs we have full access to the generating distribution.

**Weaknesses:**

The paper's failure to address several important issues, in addition to the numerous ambiguities in the exposition, diminishes the paper's overall quality. Therefore, I am inclined to reject the current version of the paper. My concerns include the following.

- **Basic MEC setup is unclear.** Throughout the paper, the authors seem to *implicitly* assume that X, Y are both vectors (or strings). In fact, the prefix tree in Section 4 does not even make sense unless elements of X are strings. This implicit vector assumption is also used in Section 2.3, Algorithm 1 and 2. However, this assumption on the structure of X and Y doesn’t seem to be stated explicitly anywhere in the paper.

    Also, what kind of access do we have to the marginals of X and Y? Do we get black-box queries to the probability mass evaluations? Is it a sampling oracle?

- **Notion of efficiency is never formally defined.** The main reason for using heuristic MEC algorithms instead of the provable approximation algorithms is to avoid the log-linear-in-N run time. Computational efficiency, at least in CS, is always be defined relative to some problem instance size (i.e., it is an asymptotic notion). If N is “too big”, then what is the right index for the instance size for MEC? This would make sense if one explicitly assumed that X is supported on {0,1}^n and Y is supported on {0,1}^m, and we call any quantity “too large” or “intractably large” if it grows superpolynomially in n or m.

- **Small support size of conditional distributions does not imply efficient sampleability (Condition 4.6).** Suppose X is a random vector in {0,1}^n defined as the output distribution of a pseudorandom generator (PRG), i.e., the pushforward of the uniform distribution over {0,1}^s through the PRG. Clearly, each conditional distribution arising in the autoregressive form of this distribution is supported on {0,1}, which is of size 2. However, these conditional distributions are not even efficiently computable (since they are given by a PRG).

- **Missing run-time analysis of subroutines.** In Algorithm 3 (IMEC), is it clear that the optimization over the collection of partitions U can be implemented efficiently?

- **Motivation for ARIMEC.** In what sense is ARIMEC better than previous approaches? In the regimes where TIMEC and FIMEC performs well, is it expected that ARIMEC performs not worse than these two approaches? Also, what is the motivation behind using partitions defined using the prefix tree?

- **Missing details on the steganography task.** Details of the steganography experiment are missing (even including Appendix D), which makes it hard to understand what experiment is exactly being conducted here. Could the authors please expand on the last paragraph of Section 5?

**Questions:**

- What kind of access do we have to the marginals of X and Y? Do we get black-box queries to the probability mass evaluations? Is it a sampling oracle?
- In Condition 2.5, what qualifies as “small” support size and what quantifies as “intractably large”?
- In Algorithm 3 (IMEC), is it clear that the optimization over the collection of partitions U can be implemented efficiently?
- If the encoding is perfectly secure (i.e., encoding of ciphertext X into stegotext S is deterministic) and the distribution of stegotext S and covertext C is the same, doesn’t this mean that one can “hallucinate” secret messages from innocuous text?

**Editorial comments**

- The NP-hard reference is rather misleading. For the NP-hardness result, the instances are indexed by N, the support size. Even if MEC were in P, this would not suffice for the setting this paper is interested in.
- In the Algorithm boxes, the subscript 1:j-1 doesn’t make sense for j = 1.

---

> ### Author Response · Authors · 2023-11-17
> **Response to Reviewer caMu**
>
> We thank the reviewer for their engaged review and helpful comments. We feel that the reviewer's feedback has helped us significantly improve the submission, of which we have uploaded a revised copy.
>
> > Basic MEC setup is unclear. Throughout the paper, the authors seem to implicitly assume that X, Y are both vectors (or strings). In fact, the prefix tree in Section 4 does not even make sense unless elements of X are strings. This implicit vector assumption is also used in Section 2.3, Algorithm 1 and 2. However, this assumption on the structure of X and Y doesn’t seem to be stated explicitly anywhere in the paper.
>
> We would like to clarify that the original submission *did explicitly state this assumption*. Specifically:
> - In Condition 2.5, it states "Y = (Y_1, . . . , Y_m) is a random vector".
> - In Condition 2.6, it states "X = (X_1, . . . , X_n) is a random vector".
> - In Condition 4.6, it states "X = (X_1, . . . , X_n) is a random vector"
>
> However, in order to address the reviewers other comments, we have removed the conditions from the submission. Thus, to make this point clear, we added explicit mention of random vectors to Section 1, Section 2.2, and Section 2.4:
> - These algorithms work by iteratively coupling components of random vectors using provable MEC approximation algorithms in such a way that guarantees the aggregate joint distribution is a coupling.
> - In some settings, it is desirable to (non-provably) approximate minimum-entropy couplings where one random vector assumes such a large number of possible outcomes that the approaches described in Section 2.2 are inapplicable. Sokota et al. (2022) propose an iterative approach to such settings, which we call TIMEC, that assumes the random vector is autoregressively specified.
> - Sokota et al. (2022) also proposed a second approach, which we call FIMEC, in which $X$ is also assumed to be a random variable
> - $X=(X_1, \dots, X_n)$ is a random vector such that $X_i$ and $X_j$ are independently distributed for $i \neq j$.
>
> Please let us know if there are other points in the submission where the reviewer feels it would be informative to add clarification about the status of $X$ and $Y$ as random vectors.
>
> > Also, what kind of access do we have to the marginals of X and Y? Do we get black-box queries to the probability mass evaluations? Is it a sampling oracle?
>
> We get access to probability mass evaluations. We have added the text below to the introduction to make this more explicit: "The problem of computing a coupling with the minimum amount of joint entropy among all feasible couplings, given access to probability mass evaluations of the marginals, is called minimum-entropy coupling (Kovacevic et al., 2015)."
>
> > Formal Notion of Efficiency / Clarity of Conditions / Runtime Analysis
>
> We thank the reviewer for this feedback. We agree that the way we were presenting the material previously was lacking formality. We attempted to address the reviewer's concerns in the revised version of the submission. To summarize:
> - We now describe algorithm runtime using big-O analysis.
> - We removed all of the "conditions" from our presentation of the material (including but not limited to those mentioned as of concern by the reviewer).
> - Regarding Algorithm 3 in its generic form, it is *not* the case that optimization over the collection of partitions U can be implemented efficiently. To make this more clear, we have added the text: "Note that whether Algorithm 3 can be implemented efficiently depends on the distribution $\nu$ and the set of partitions $\mathfrak{U}$."
> - Following up on the previous point:
>     1. For TIMEC, optimization over the collection of partitions can be implemented efficiently (the finest partition will always have maximum entropy).
>     2. For FIMEC, optimization over the collection partitions can be implemented efficiently under the assumption that the distribution is factorable.
>     3. For ARIMEC, it is less obvious that optimization over the collection of partitions can be implemented efficiently. To address this issue, we introduced a pruning technique (which was in the appendix of the original submission) that can prove large parts of the prefix partition tree cannot induce a maximum-entropy partition. While we do not show a polynomial time guarantee for this pruning procedure, we found that, for all of the distributions that we have tried in the experiments, it only required checking a small number of partitions (on the order of 1) on average in order to find a provably maximum-entropy partition.

---

> ### Author Response · Authors · 2023-11-17
> **Response to Reviewer caMu (Part 2)**
>
> > Motivation for ARIMEC
>
> We attempt to make the motivation clear in the abstract: "We derive a new IMEC instance from this formalism, which we call ARIMEC, that, unlike existing IMEC algorithms, can be applied in practice to arbitrary discrete distributions, and introduce associated operations that make ARIMEC efficient in practice." To state in different words, ARIMEC is the first algorithm that can produce couplings of large support distributions of arbitrary structure in practice. This contrasts with FIMEC, which requires one of the two distributions to be factorable. Please let us know if there is a better way we can characterize this motivation.
>
> > Missing details on the steganography task. Details of the steganography experiment are missing (even including Appendix D), which makes it hard to understand what experiment is exactly being conducted here. Could the authors please expand on the last paragraph of Section 5?
>
> We apologize that the details here were unclear. To clarify:
> 1. For the information-theoretic steganography experiments, the task is to perform a coupling between uniformly randomly distributed byte strings and the distribution over 100 tokens induced by GPT-2. The error rate is the proportion of the time that the MAP byte string is not equal to the true byte string. Symbolically $E_{X} E_{Y \sim \gamma(Y \mid X)} I[X \neq \arg \max_{x} \gamma(x \mid Y)]$. We have added this clarification about error rate to the main text.
> 2. For unencrypted steganography, the task is to perform a coupling between the distribution over 100 tokens induced by GPT-2 conditioned on two different prompts. Error rate is defined the same way.
>
> Please let us know if there is any additional detail that would be helpful to add.
>
> > If the encoding is perfectly secure (i.e., encoding of ciphertext X into stegotext S is deterministic) and the distribution of stegotext S and covertext C is the same, doesn’t this mean that one can “hallucinate” secret messages from innocuous text?
>
> Naively, yes, if the receiver is expecting a message when no message is actually being sent, this is exactly what will occur. However, in practice, one can mitigate this possibility using standard error detection techniques. For this reason, the information-theoretic steganography model introduced by Cachin (1998), which we used in the submission, is common in steganography literature.
>
> > In the Algorithm boxes, the subscript 1:j-1 doesn’t make sense for j = 1.
>
> We use upper bound inclusive indexing, so 1:j-1 for j=1 (i.e., 1:0) means the empty array (as intended). We added this note to the submission to clarify this point: "Note that we use upper-bound-inclusive indexing, so $Y_{1:0}=()$, $Y_{1:1}=(Y_1)$, $Y_{1:2}=(Y_1, Y_2)$, etc."
>
> ---
>
> We thank the reviewer again for their feedback. Please let us know if you have any further suggestions for the revised text.

---

> ### Author Response · Authors · 2023-11-22
> **Follow up**
>
> Hello Reviewer caMu, we would be grateful if you can confirm whether our revision has addressed your concerns, and let us know if any issues remain. To recap, we:
> - Revised our discussion of efficiency and provided runtime analysis
> - Clarified concerns regards assumptions & motivation
> - Answered additional questions
>
> We hope the reviewer will be able to respond before the end of the discussion (November 22nd).

---

> ### Comment · Reviewer_caMu · 2023-11-23
>
> I greatly appreciate the responsiveness of the authors. The revisions, especially their new run-time analysis, have clarified the setup and motivation of the proposed approach. However, the run-time analysis for ARIMEC is still missing a few important details, and could benefit from further investigation.
>
> - **Computing the entropy for prefix tree partitions.** Apart from taking the maximum over the collection of partitions $\mathfrak{U}$, Is it clear that for **each** prefix tree partition $P_v$ the entropy of the conditional distribution $B_v \mid y_{1:j-1}$ can be efficiently computed? This seems to require **additional conditions on the marginal $\mu$ of $X$**. I would assume that $\mu$ should be able to efficiently compute the (conditional) probability mass of any subtree in $\mathbb{X}$. This is satisfied if $\mu$ is a product distribution (Assumption 2.4), but it’s not clear what weaker assumptions on $\mu$ would ensure this.
> - **Conditions on the marginal $\nu$ of $Y$.** This is similar to the issue mentioned above. All heuristic coupling algorithms presented here require efficient computation of the conditional distributions $\nu(Y_j \mid Y_{1:j-1})$ for any j. The implicit assumption seems to be that $\nu(Y_1,…,Y_{j-1})$ is efficiently computable. The issue here is that $Y_1,…,Y_{j-1}$ are not elements of $\mathbb{Y}$. They are prefixes (i.e., subsets) whose cardinality (in terms of elements of $\mathbb{Y}$) may be exponentially large in m. Thus, the sentence “given access to the probability mass evaluations of the marginals” in p.1 needs further specification. Viewing $\nu$ (and also $\mu$) as an oracle, what queries are allowed in the paper’s setup?
> - **The number of nodes visited in the prefix tree (Z).** The current run-time analysis of ARIMEC (Proposition 4.1) could be investigated further. The introduction of the parameter Z, the number of nodes visited in the prefix tree, is rather ad hoc. This is understandable given the rebuttal time constraints, but it glances away from potentially important details of ARIMEC.

---

> > ### Author Response · Authors · 2023-11-23
> > **Response**
> >
> > > Computing the entropy for prefix tree partitions
> >
> > Yes, for ARIMEC, this can be efficiently computed given access to $\mu(X_i \mid X_{1:i-1})$. This is discussed in Proposition B.1.
> >
> > > All heuristic coupling algorithms presented here require efficient computation of the conditional distributions $\nu(Y_j \mid Y_{1:j=1})$ for any $j$.
> >
> > Yes. Note that we explicitly made this assumption in the original submission and that it is explicitly stated here: "Sokota et al. (2022) propose an iterative approach to such settings, which we call TIMEC, that assumes the random vector is autoregressively specified."

---

> ### Comment · Reviewer_caMu · 2023-11-23
>
> Thank you for the clarification. It's not immediately clear to me how Proposition B.1 explains the posterior update procedure. Let's consider a simple example where $\mathbb{X} = \\{0,1\\}^n$ and $\mathbb{Y}=\\{0,1\\}^m$. Suppose in iteration $j=1$, the partition used is $\Pi_1 = \\{\\{0^n\\},\mathbb{X}\setminus\\{0^n\\}\\}$. At the beginning of iteration 2, what is the measure of $\gamma(0*\mid Y_1=0)$ (i.e, set of all strings with prefix $X_1 = 0$) in terms of $\\{\gamma(B,Y_1)\mid B \in \Pi_1\\}$ and $\mu$?
>
> More explanation on the posterior update, possibly using simple examples like this, would be helpful.
>
> On the other hand, the assumption on the marginal $\nu$ of $Y$ is clear now. A minor nitpick is whether "random vector is autoregressively specified" is a standard way of describing such measures. I would instead write "the measure $\nu$ efficiently computes prefix queries" or "measures of cylinder sets", but I acknowledge that different communities might describe it differently.

---

> ### Author Response · Authors · 2023-11-23
> **Clarification of Posterior Update Procedure**
>
> Proposition B.1 shows how, given the posterior associated to a partition $P_v$, the partition associated to a partition $P_u$ can be computed, where $u$ and $v$ are neighbors in the prefix tree. In the example you describe, $\emptyset$ and $0^n$ are $n$ edges away from one another in the prefix tree, so computing the posterior over $\gamma(0^{\ast} \mid Y_1 = 0)=\gamma(\mathbb{B}_{\sqsubset 0} \mid Y_1=0)$ would require performing $n$ updates as dictated by Proposition B.1.
>
> To give intuition, Proposition B.1 shows that the posteriors of adjacent nodes are related by up-weighting or down-weighting the value of the appropriate edge. For example, if we perform a coupling on the partition $P_{00}$ and the posterior
>
> $\gamma(\mathbb{B}_{\not \sqsubset 00} \mid Y_1)$
>
> is higher than the prior
>
> $\gamma(\mathbb{B}_{\not\sqsubset 00})$
>
> then the posterior associated with $P_0$ given $Y_1$ is a re-weighting of the prior where the probability
> $\gamma(\mathbb{B}_{\sqsubset 00} \mid Y_1)$ is shifted downward and the probability of all other blocks are shifted upward (as per Proposition B.1).
>
> We will add detailed examples on this point to the text. Please let us know if any other details would be helpful. We also commit to release well-documented code for all of the algorithms discussed in the paper (so these procedures will also be given explicitly).

---

### Author Response · Authors · 2023-11-20
**General Comment**

We thank all of the reviewers again for their constructive reviews. In response to the reviews, we made substantial changes to the presentation of the submission. We believe that these changes address the concerns raised in the initial reviews. As the rebuttal period ends soon, we would appreciate if the reviewers could update their evaluations and scores, clearly articulating any aspect they still don't like about the paper so that we can continue improving it for the future.

---

### Meta-Review · Area_Chair_JkwH · 2023-12-06

**Metareview:**

The paper proposes a new framework for heuristic iterative minimum-entropy coupling (IMEC) algorithms capturing two previous algorithms (TIMEC, FIMEC) and a new one (ARIMEC). Minimum entropy coupling of two distributions can be approximated in log-linear time in the size of the support. However, the paper focuses on the setting where the domain is huge and aims for runtime polynomial in the logarithm of the support size. Previous heuristics have been developed for the settings where one of the distribution is factorable into many independent factors. The new algorithm (ARIMEC) can be applied to arbitrary discrete distributions.

On the downside, the new algorithm is not generally efficient and it depends on the particular instance/family of distributions whether it can be implemented efficiently. The paper shows the utility of the method on the applications in the previous work, namely Markov coding games and steganography. The reviewers note that these applications are not commonly discussed and would want a better idea of the general applicability of the new algorithm. One issue noted by several reviewers is on the writing quality of the paper, especially that the model, the algorithm, and the theorem statements are not sufficiently formal, leading to confusion about how the distributions are accessed, the assumptions, and the desired runtime.

**Justification For Why Not Higher Score:**

Two reviewers recommend rejection and one puts it slightly above the threshold. The reviewers generally complaint about the writing quality and one reviewer is concerned with the limited application beyond the original applications, despite the more general algorithm.

**Justification For Why Not Lower Score:**

N/A

---

### Decision · Program_Chairs · 2024-01-16

Reject